# FOXA1 loss drives basal/squamous de-differentiation of prostate cancer and induces an immunosuppressive tumor microenvironment

Lourdes Brea [1,2], Hongshun Shi[1], Viriya Keo[1,2], Jing Huang [3], Liu Peng[1], Qi Chu[1], Wanqing Xie[1], Yinghua Xie[1], Sambhavi Senthil[1], Matthew T. Breneman[4,5], Jie Fan[2], Ping Xie[2], Xiaodong Lu [1], David J. Degraff[6], Sarki A. Abdulkadir [7], Ximing Yang[8], David Kosoff[4,5], Jonathan C. Zhao [1,3,9], Bin Zhang [2], Jian Hu [3] & Jindan Yu [1,3,9] ✉

FOXA1 is a prostate lineage-specifying transcription factor that is frequently dysregulated or mutated in prostate cancer (PCa). While FOXA1 has been reported to exhibit both PCa-promoting and -inhibitory functions, its role within an immune-proficient PCa context remains unclear. Here, we show that prostate-specific deletion of *Foxa1* in *Pten*-deficient mice drives tumor progression by reprogramming luminal PCa cells toward a basal/squamous-like state and promoting an immunosuppressive tumor microenvironment. Histological and transcriptomic analyses reveal aggressive tumors with extensive basal/squamous features, a reactive stroma, and disorganized tissue architecture. Mechanistically, FOXA1 directly represses basal/squamous and inflammatory genes, which become activated upon its depletion. This is accompanied by an accumulation of immunosuppressive myeloid cells, dysfunctional T cells, and immunosuppressive cytokine signaling. Together, these findings demonstrate a tumor-suppressive role for FOXA1 as an enforcer of luminal identity, such that its loss drives basal/squamous de-differentiation, inflammatory response, and immunosuppression.

Forkhead box A1 (FOXA1) is a master regulator of utmost importance in prostate cancer (PCa) progression and is one of the most frequently mutated genes in PCa[1-4]. It is a pioneering factor which critically recruits the androgen receptor (AR) to lineage-specific enhancers to induce prostatic gene expression[5-7]. FOXA1 is initially upregulated in primary prostate cancer (PCa) vs. benign prostate, but is ultimately downregulated in late-stage castration-resistant PCa (CRPC), wherein AR, in contrast, is amplified and overexpressed[8-12]. In this low milieu of FOXA1, AR has been shown to become unregulated by FOXA1 and thus gain broad binding events, some of which are at oncogenic sites[6]. In addition, FOXA1 loss further induces epithelial-to-mesenchymal transition (EMT) and neuroendocrine differentiation, at least in part, via

[1]Department of Urology, Emory University School of Medicine, Atlanta, GA, USA. [2]Division of Hematology/Oncology, Department of Medicine, Northwestern University, Chicago, IL, USA. [3]Department of Human Genetics, Emory University School of Medicine, Atlanta, GA, USA. [4]William S. Middleton Memorial Veterans' Hospital, Madison, WI, USA. [5]Carbone Cancer Center, University of Wisconsin-Madison, Madison, WI, USA. [6]Department of Pathology and Laboratory Medicine, Pennsylvania State University College of Medicine, Hershey, PA, USA. [7]Department of Urology, Northwestern University Feinberg School of Medicine, Chicago, IL, USA. [8]Department of Pathology, Northwestern University Feinberg School of Medicine, Chicago, IL, USA. [9]Winship Cancer Institute, Emory University School of Medicine, Atlanta, GA, USA. ✉e-mail: jindan.yu@emory.edu

upregulating cytokines, such as TGFB3 and IL8[9–11,13]. Accordingly, FOXA1 has shown context-dependent anti- or pro-tumor roles as its depletion inhibits growth in hormone-sensitive PCa[3,10] but induces aggressive phenotypes in CRPC[6,10]. Moreover, FOXA1 harbors a wide spectrum of mutations that also show distinct functional consequences, including pro- or anti-luminal differentiation and pro- or anti-tumor progression[2,3]. Recently, FOXA1 loss was shown to induce macrophage infiltration in an in vitro co-culture system under hypoxic conditions mimicking the tumor environment[14]. In contrast, using cell culture and subcutaneous allograft models treated with IFN inducers, such as poly(I:C), He et al. reported increased IFN response and T cell infiltration in *FOXA1*-knockdown cells[15]. However, a comprehensive analysis of FOXA1 in an immune-competent and autochthonous PCa setting is still lacking.

In recent years, many previously hard-to-treat or treatment-resistant cancer types, including melanoma and lung cancers, have benefited greatly from immunotherapies, such as immune checkpoint inhibitors (ICIs). Unfortunately, PCa patients have shown an overall poor response to ICIs due to the immune-cold nature of the disease[16–19]. Recent studies have demonstrated that PCa is highly infiltrated by T regulatory cells (Tregs), myeloid-derived suppressor cells (MDSCs), and tumor-associated macrophages (TAMs) that promote immunosuppression and enable tumor progression[20,21]. Moreover, metastatic CRPC, which has increased lineage plasticity, has been shown to exhibit a particularly immune-suppressive tumor immune microenvironment (TIME), with increased TGF-β signaling and increased abundance of T helper 17 (Th17) cells, exhausted cytotoxic T lymphocytes (CTLs), and tumor-promoting myeloid populations, such as M2-macrophages and inflammatory monocytes[22–25].

Emerging evidence suggests that certain genetic and epigenetic traits of PCa critically regulate the composition of tumor immune infiltrates in its TIME[26–30]. For example, several studies have explored the effects of *PTEN* loss, a frequently altered tumor suppressor gene that is inactivated in ~20-30% of primary PCa and up to ~60% of CRPC[12,31,32]. Notably, prostate-specific knockout (KO) of *Pten* in mice, which leads to PCa initiation, induced the expression of immunosuppressive cytokines CXCL1, CXCL2, IL10 and IL6 via JAK/STAT3 signaling, leading to increased prostate tumor infiltration by MDSCs and decreased accumulation of T cells[26,33,34]. Moreover, other genetic aberrations, such as *Trp53* KO, *Arid1a* KO, *Zbtb7a* KO, and *Pml* KO, have also been shown to alter the TIME of PCa[26,35]. However, it remains unclear whether there are prostate lineage-defining factors that uniquely predispose PCa to an immunosuppressive TIME.

Here, we generated a genetically engineered mouse model with prostate-specific deletion of *Foxa1* in the *Pten*-null mice. Combining mouse histology with single-cell and spatial transcriptomics and Cytek spectral flow analysis, we report that *Foxa1* loss induces luminal-to-basal/squamous lineage switch and remodels the TIME in PCa, promoting tumor infiltration by immunosuppressive myeloid cells and dysfunctional T cells, which was validated in human PCa. We present FOXA1 as a master regulator of luminal cell lineage and the PCa immune landscape.

## Results

### Depletion of *Foxa1* in PCa epithelial cells leads to basal/squamous de-differentiation, a loss of prostate tissue architecture, and a reactive stroma

Prostate-specific deletion of *Foxa1* in the PbCre4:*Foxa1*f/f mouse (denoted as F mouse), which places the Cre recombinase gene under the control of the Pbsn gene promoter and directs Cre expression to prostate epithelial cells[36], has been shown to induce prostatic hyperplasia, but not tumorigenesis[37]. To examine FOXA1 function in the context of PCa, we crossed the F mouse with the *Pten*-null transgenic mouse (P mouse), a well-established model that develops PCa, representing a large number of human PCa with *PTEN* loss[32], and generated

the PbCre4:*Pten*f/f*Foxa1*f/f (PF mouse) model with prostate-specific deletion of both *Pten* and *Foxa1* genes. As mice are known to naturally exhibit low androgen levels, closer to those observed in patients undergoing standard androgen deprivation therapy (ADT)[38], the PF tumors arise in an androgen-low environment that parallels CRPC, a setting in which *FOXA1* is commonly downregulated[8–11], often co-occurring with *PTEN* loss (Figure S1A). Genotyping PF mice using allele-specific primers confirmed the presence of floxed *Foxa1* and *Pten* alleles, whereas immunohistochemistry (IHC) staining validated the depletion of PTEN and/or FOXA1 proteins in the respective mice, compared to the PbCre4-negative wild type (WT) mice (Fig. 1A-B).

To understand the tumorigenic effects of the genetic deletions in P vs. PF mice, we first examined the prostates of 15-week (wk)-old mice by hematoxylin and eosin (H&E) staining (Fig. 1C). *Pten*-null prostates exhibited cribriform morphology and intraductal carcinoma, being consistent with previous reports[32,39]. Notably, concurrent deletion of *Foxa1* in the context of *Pten* KO in the PF mice led to dramatic alterations to all lobes (AP, DLP, and VP) of the prostate (Fig. 1C & S1B). Compared to the typical glandular architecture of PCa observed in the P mice, the tumors of the PF mice exhibited remarkably disordered cell arrangements with a loss of glandular architecture, suggesting more aggressive behavior. Even as early as 9 weeks of age, the PF prostates exhibited a largely disorganized and aggressive phenotype compared to P mice (Figure S1C). Further evaluating PF mice across different timepoints, we observed that PF prostate tumors exhibited abundant evidence of PCa invasion, necrosis, and a reactive stroma, characterized by extracellular matrix remodeling and inflammation[40] (Fig. 1D & S1C).

Interestingly, PF tumors also exhibited a massive gain in adeno-squamous differentiation with keratinization, displaying large keratin pools (Fig. 1D & S1D), suggesting that *Foxa1* deletion in the prostate leads to epithelial cell de-differentiation and induces lineage plasticity. Indeed, IHC staining of 18-wk-old prostates revealed a widespread gain in the expression of basal marker p63 in tumor epithelial cells in the PF mice compared to the P mice, accompanied by partial loss of luminal marker AR and KRT8 (Fig. 1E). Notably, PF prostate epithelial cells also exhibited increased staining for the proliferation marker Ki67, suggesting increased proliferation with *Foxa1* loss. Lastly, in line with our findings of poorly differentiated and highly proliferative tumors in the PF prostates, we observed that the survival of the PF mice was greatly shortened as compared to P mice (Fig. 1F). Further investigation of the anatomy of the PF mice revealed that they likely died from bladder outlet obstruction, a common complication of locally advanced prostate cancer also found in humans[41].

To further investigate how *Foxa1* loss influences tumor progression and phenotype under a more advanced, profoundly androgen-depleted CRPC-like context, P and PF mice were castrated at 12–18 weeks old and harvested 3–12 months later. Castrated P and PF mice are referred to as XP and XPF, moving forward. H&E and IHC staining for Ki67 in XP and XPF prostate tissues demonstrated reduced epithelial cell growth and proliferation in XP mice as compared to XPF, as seen in intact mice. Meanwhile, XPF prostate tissues exhibited a disordered and poorly differentiated tumor phenotype (Fig. 1G & H, Figure S1E), with a loss of luminal identity (indicated by reduced AR and KRT8 staining), and widespread gain in basal phenotype (indicated by p63 staining). Moreover, XPF tumors were found to be significantly larger than XP tumors post castration, suggesting some survival advantage of *Foxa1* loss (Figure S1F). Histopathological evaluation of tissue sections from lymph nodes, lung, and/or liver of 9 intact PF mice, 7 intact P mice, and 5 castrated XP mice revealed no evidence of metastasis, while 2 out of 5 evaluated XPF mice exhibited evidence of metastasis at distant organs (Figure S1G). Thus, it appears *Foxa1* loss promotes aggressive castration-resistant disease in our genetically engineered mouse models. Altogether, our histopathological analysis of the P and PF mice tumors found that *Foxa1* loss led to luminal cell de-differentiation, a loss of prostate glandular architecture, and

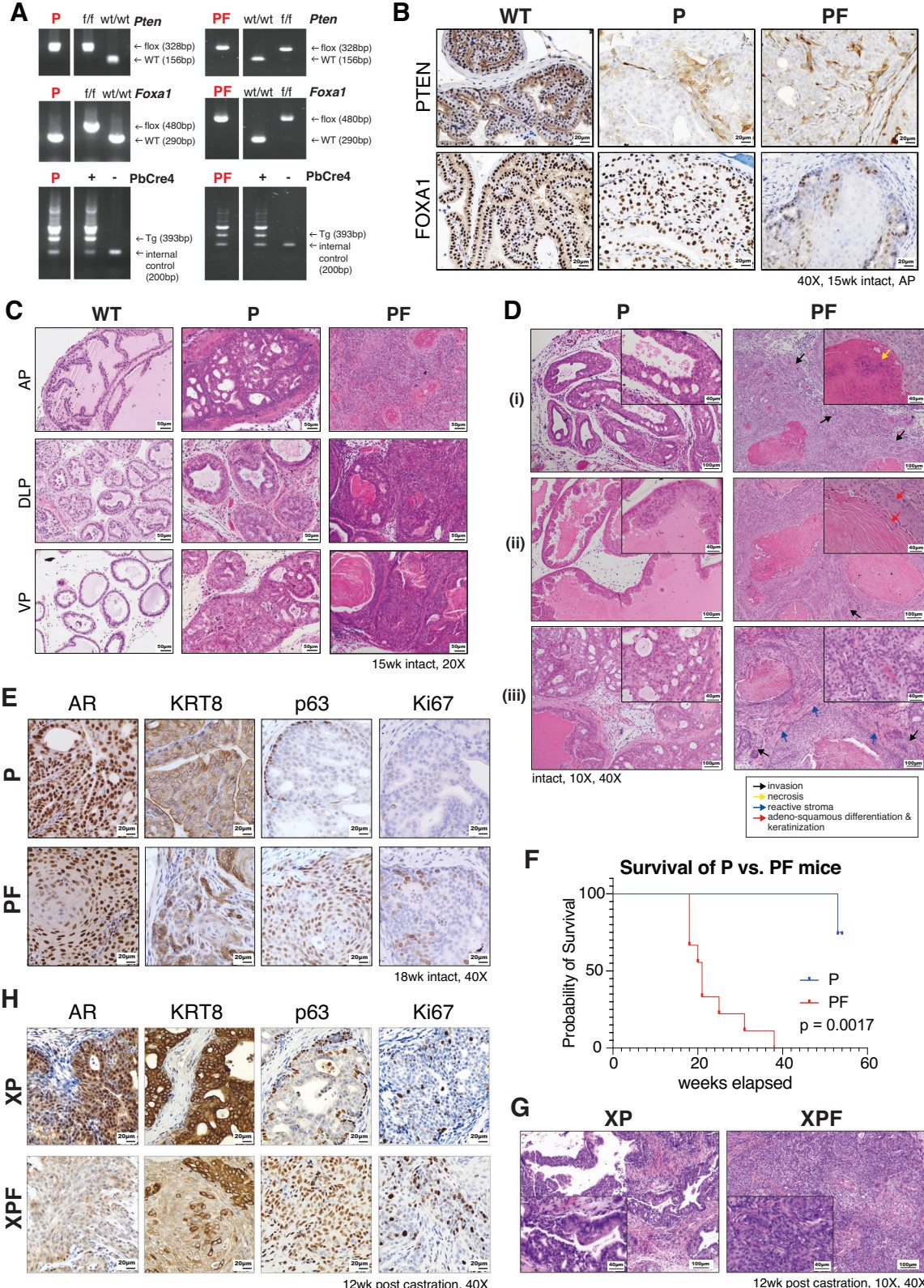

40X, 15wk intact, AP

15wk intact, 20X

intact, 10X, 40X

→ invasion
→ necrosis
→ reactive stroma
→ adeno-squamous differentiation & keratinization

18wk intact, 40X

12wk post castration, 40X

**Survival of P vs. PF mice**

p = 0.0017

12wk post castration, 10X, 40X

aggressive epithelial growth patterns, which are further exacerbated by castration.

### *Foxa1* loss in PCa cells induces lineage plasticity and a basal/squamous cell population

To understand the molecular basis underlying the marked histological changes of *Foxa1*-depleted PCa, we performed 10x Genomics scRNA-seq analysis of whole prostates dissected from the P and PF mice at an early 12wk (P12 vs. PF12) and a late 18wk (P18 vs. PF18) timepoints, as well as mice castrated at 18wk and harvested 12wk post castration (XP vs. XPF). Clustering of all single cells using the Louvain algorithm and Uniform Manifold Approximation and Projection (UMAP) visualization resulted in 8 major cell type clusters: *Krt8*-high luminal-like epithelial cells, *Krt5*-high basal-like epithelial cells, stromal cells, monocytes,

**Fig. 1 | *Foxa1* depletion in PCa epithelial cells leads to a more aggressive disease phenotype with basal/squamous differentiation and loss of glandular architecture. A** Example of genotyping results for PbCre4:*Pten^{f/f}* (P) and PbCre4:*Pten^{f/f}Foxa1^{f/f}* (PF) mice, along with expected band sizes for *Pten/Foxa1* flox (f/f) or WT (wt/wt) alleles, and PbCre4 status (Tg = transgene). Genotyping was performed routinely for colony maintenance, and representative results are shown. **B** IHC staining of PbCre4-negative (WT), PbCre4:*Pten^{f/f}* (P), and PbCre4:*Pten^{f/f}Foxa1^{f/f}* (PF) mouse prostate tissue sections (40X, 15wk, AP lobe) with antibodies against PTEN and FOXA1 proteins. Similar results were observed across ≥6 independent biological replicates for P and PF mice. **C** H&E staining of AP (anterior), DLP (dorsal-lateral), and VP (ventral) lobes for 15wk-old intact WT, F, P and PF mouse prostates (20X). Similar histological patterns were observed in ≥8 mice per genotype analyzed across multiple time points. **D** Representative H&E images of age-matched (i) 12wk DLP, (ii) 18wk VP, and (iii) 18wk AP lobes of intact P vs. PF prostate tissue samples demonstrating evidence of PCa invasion (indicated by black arrows), necrosis (yellow arrow), adeno-squamous differentiation and keratinization (red arrows), and reactive stroma (blue arrows) in PF mice. Similar histological patterns were observed in ≥ 8 mice per genotype analyzed across multiple time points. **E** IHC staining of 18-wk intact P vs. PF prostates (40X) for AR, KRT8 (luminal marker), p63 (basal marker), and Ki67. Similar results were observed in ≥ 3 independent biological replicates per genotype spanning 12–18 weeks of age. **F** Kaplan-Meier survival curve for P vs. PF mice (P *n* = 4, PF *n* = 9, *p*-value = 0.0017, log rank Mantel-Cox test). Source data are provided in the Source Data file. **G** Representative H&E images of castrated PbCre4:*Pten^{f/f}* (XP) and PbCre4:*Pten^{f/f}Foxa1^{f/f}* (XPF) mice mouse prostates (10X, 40X). Similar histological findings were observed across all analyzed mice (XP, *n* = 6; XPF, *n* = 5). **H** IHC staining of castrated XP vs. XPF prostates (40X) for AR, KRT8 (luminal marker), p63 (basal marker), and Ki67. Similar results were observed in 3 independent biological replicates per genotype.

macrophages, B cells, T/NK cells, and *Pate4*-high cells of seminal vesicles that are likely due to their incomplete separation from the prostates (Fig. 2A). Of these, *Epcam*+ prostate epithelial cells, including luminal-like and basal-like clusters, comprised the majority of cells of the prostate (Fig. 2A, B). Within the luminal-like cluster, we further identified four subclusters, consisting of *Wfdc2*-high, *Pbsn*-high, *Spink1*-high, and *Tff3*-high populations, whereas there were three *Trp63*-high (basal), *Mki67*-high (proliferative), or *Krt6a*-high (squamous) subclusters within the basal-like cluster (Figure S2A–D).

A comparison of cells derived from the P vs PF mice revealed a strong increase of basal-like epithelial cells in the PF prostates, which was also accompanied by an increase in immune cells (especially myeloid cells) (Fig. 2C, D). Visualization of gene expression intensity on the UMAP confirmed a drastic increase of *Krt5* (basal) and *Krt6a* (squamous) expression in epithelial cells derived from PF12 and PF18 prostates compared to P12 and P18, while the expression of luminal marker *Krt8* was not substantially affected (Fig. 2E). XPF prostate epithelial cells also exhibited a dramatic increase in basal/squamous phenotype (Fig. 2D & S2E).To further validate this, we quantified the major luminal (*Foxa1, Ar, Krt8, Krt18, Cd24a*), basal (*Trp63, Krt5, Krt14, Lgals7*), and squamous (*Krt6a, Krt13, Krt16*) marker gene expression within all epithelial cells of each mouse. Our data showed that basal/squamous gene expression was markedly upregulated by *Foxa1* KO across both intact and castrated conditions, while luminal genes were only slightly reduced in PF compared to P prostate epithelial cells (Fig. 2F). Additionally, we found increased expression of EMT genes (*Vim, Snai2, Twist1*) upon *Foxa1* deletion, especially under castrated conditions, suggesting increased invasive potential (Figure S2F).

To further confirm that *Foxa1* loss induces basal/squamous de-differentiation, we performed trajectory analyses of epithelial cells within all samples and created a joint pseudotime (Figure S2G). Plotting key luminal (*Krt8*), squamous (*Krt6a*), and basal (*Krt14, Krt5, Trp63*) genes along the joint pseudotime, we observed marked differences in their expression across the luminal-to-basal trajectory in P vs. PF samples (Fig. 2G). For instance, PF18 tumors appear to exhibit a much earlier gain in basal gene expression along the trajectory, with a lot more cells within the "early" luminal populations expressing basal markers. PF tumors also harbor a more prominent transitionary (basal/squamous) population and have a major increase of "final" basal population, as compared to P18 tumors. A very similar pattern was also observed for PF12 vs. P12 and XPF vs. XP epithelial cells on trajectory analyses (Figure S2H, I). Altogether, these results suggest that FOXA1 is required to suppress basal/squamous and EMT gene expression in PCa cells and that *Foxa1* loss leads to lineage plasticity.

## FOXA1 binds the chromatin to directly induce luminal genes, but repress basal/squamous and inflammatory genes

To gain further insights into the dysregulated signaling pathways underlying this observed basal/squamous de-differentiation in PF vs. P

prostates, we further analyzed the scRNA-seq data solely focusing on prostatic epithelial cells. Using a two-fold cutoff, we identified 676 and 587 genes that are respectively up- and down-regulated in the epithelial cells of PF12 vs. P12 mice and 635 and 786 genes that are respectively up- and down-regulated in the epithelial cells of PF18 vs. P18 mice. Gene Ontology (GO) analyses[42] of Hallmark gene sets revealed that PF12- and PF18-upregulated genes were strongly enriched for inflammatory pathways, cytokine signaling, EMT, and hypoxia (Fig. 3A), being consistent with the observed highly reactive stroma in the PF prostates and previous reports of FOXA1 in suppressing EMT and hypoxia in human PCa[10,13,14]. On the other hand, genes down-regulated in the epithelial cells of PF prostates were enriched for hormonal response, fatty acid metabolism, and adipogenesis, concordant with the role of FOXA1 as a pioneer factor of AR, which regulates androgen response and various aspects of cellular metabolism[6,7] (Fig. 3B). In agreement, GO analyses of Biological Processes (BP) revealed that PF12- and PF18-upregulated genes were highly enriched for BP pathways related to Skin Development, Keratinocyte Differentiation, and Cell/Tissue Migration, whereas PF12/PF18-down-regulated genes significantly overlap with genes involved in various metabolic pathways (Figure S3A, B).

As FOXA1 is a transcription factor that directly binds to DNA through consensus FKHD motif, we next attempted to address whether FOXA1 directly regulates genes of the above-noted pathways. We performed FOXA1 ChIP-seq in three biological replicates of whole prostates from age-matched 18-wk-old P and PF mice. ChIP-seq analysis identified a total of 10,541 consensus FOXA1 binding peaks on the chromatin. Motif analysis confirmed FKHD as the most highly enriched motif at FOXA1 binding sites. As expected, almost all of the FOXA1 binding events were lost in the PF mice (Fig. 3C, D). Next, to further understand the function of the genes that are bound by FOXA1, we assigned each FOXA1 binding event to the nearest gene and performed GO analysis. We found significant enrichment of androgen response genes among FOXA1-bound genes (Fig. 3E), in line with the well-established role of FOXA1 as an AR co-factor in gene activation[5]. Notably, multiple pathways related to cytokine signaling and inflammation, such as Inflammatory Response, TNFα signaling, and IL2-STAT5 Signaling, were also strongly enriched for FOXA1 binding. Together with scRNA-seq data showing upregulated expression of these pathway genes in PF epithelial cells (Fig. 3A), this data suggests that they may be directly repressed by FOXA1. Indeed, genome browser track view of ChIP-seq data demonstrated FOXA1 binding at enhancers proximal to luminal genes (*Krt8, Krt18*) in the P mice, which was abolished in the PF mice, accompanied by a reduction of H3K27ac at the target gene promoters (Fig. 3F & S3C), suggesting their being FOXA1-induced genes. Importantly, strong FOXA1 binding events were also detected at regulatory elements near basal/squamous genes (*Krt6a, Trp63*) and inflammatory genes (*Hif1a, Tnfaip2, Ifitm2/3/6*) in the P mice, which was also lost in the PF mice (Fig. 3G, H & S3D, E).

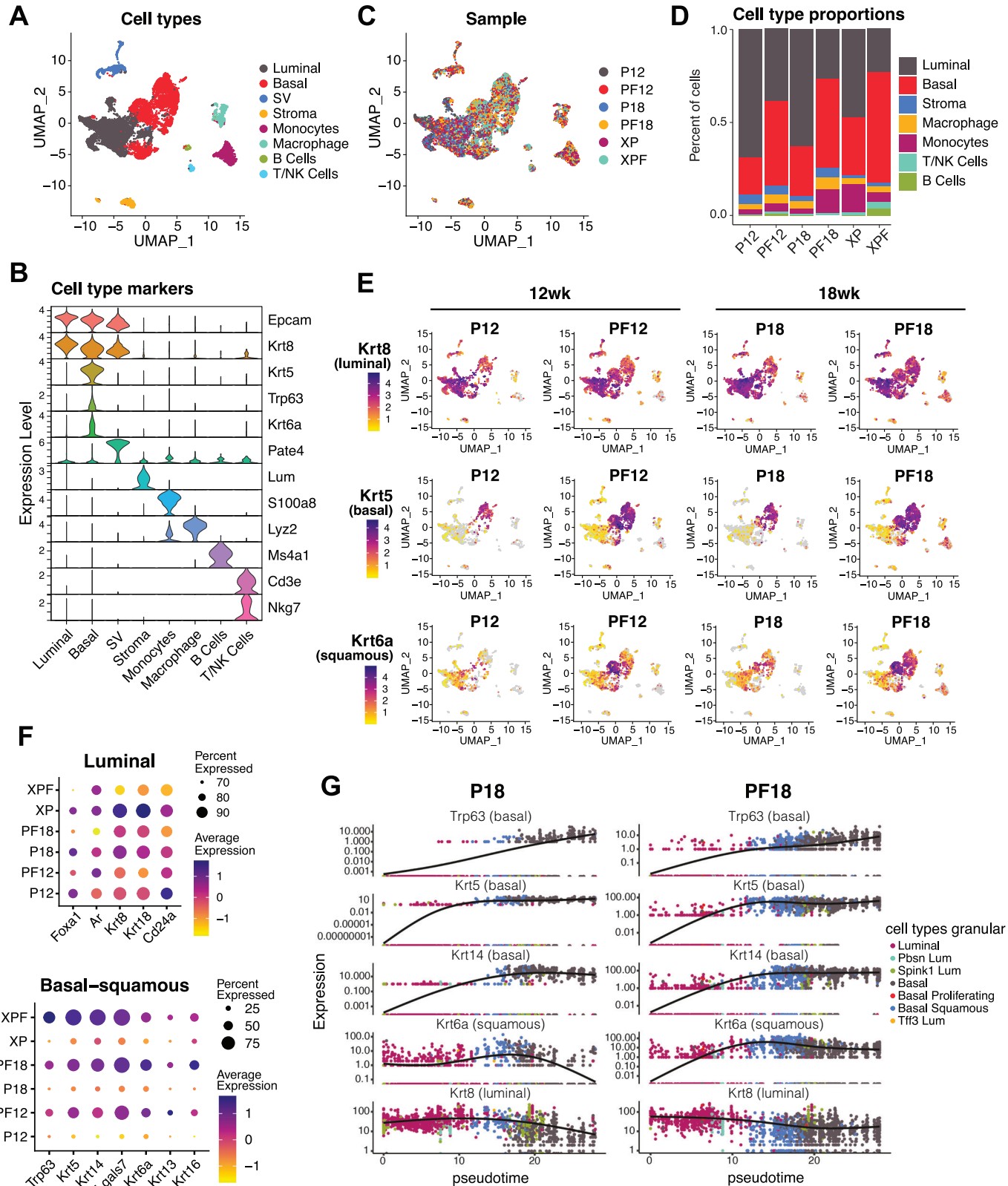

However, H3K27ac at the promoters or enhancers of these genes was increased, supporting their being FOXA1-repressed genes. This data supports that FOXA1 directly binds luminal gene enhancers to promote their expression, while it binds at basal/squamous and inflammatory genes to repress their expression.

To further validate this in human PCa, we evaluated previously published gene expression microarray analysis of human PCa LNCaP cells with *FOXA1* knockdown (KD) and rescue with WT or mutant FOXA1 (G87R, D226N, M253K, 254-257FENG > C(del), F266S, L388M)[13]. Interestingly, we found that *FOXA1* KD leads to an increase in *KRT14* and *KRT6A* expression. Rescue with WT FOXA1 subsequently repressed *KRT14* and *KRT6A*, while rescue with FOXA1 forkhead domain mutants (226, 253, 254-7FENG, and 266), which have been shown to exhibit impaired chromatin binding, had reduced ability to repress *KRT14* and

**Fig. 2 | *Foxa1* loss in PCa cells induces lineage plasticity and basal/squamous cell population. A** UMAP of integrated P12 (12wk PbCre4:*Pten^f/f*, *n* = 2263 cells), PF12 (12wk PbCre4:*Pten^f/f Foxa1^f/f*, *n* = 3169 cells), P18 (18wk PbCre4:*Pten^f/f*, *n* = 2738 cells), PF18 (18wk PbCre4:*Pten^f/f Foxa1^f/f*, *n* = 3584 cells), XP (castrated PbCre4:*Pten^f/f*, *n* = 3590 cells), and XPF (castrated PbCre4:*Pten^f/f Foxa1^f/f*, *n* = 3918 cells) 10x Genomics scRNA-seq samples with cell type clusters annotated. Coarse cell typing depicted on UMAPs shows major epithelial, immune, and stroma cell types. **B** Violin plots depicting major cell type marker genes across cell type clusters. **C** UMAP colored by mouse condition. One mouse prostate tumor was analyzed per condition. **D** Proportion of cell types in P12, PF12, P18, PF18, XP, and XPF prostate tissues. **E** Feature plots depicting *Krt8* (luminal), *Krt5* (basal), and *Krt6a* (squamous) gene expression on the integrated UMAP split by each condition. **F** Dot plot quantifications for major luminal and basal/squamous genes within epithelial populations of P12, PF12, P18, PF18, XP, and XPF samples. **G** Trajectory pseudotime analysis of epithelial cells within P18 vs. PF18 samples, performed using Monocle3. Luminal (*Krt8*), basal (*Krt5, Trp63, Krt14*), and squamous (*Krt6a*) gene expression are plotted across the joint pseudotime trajectory.

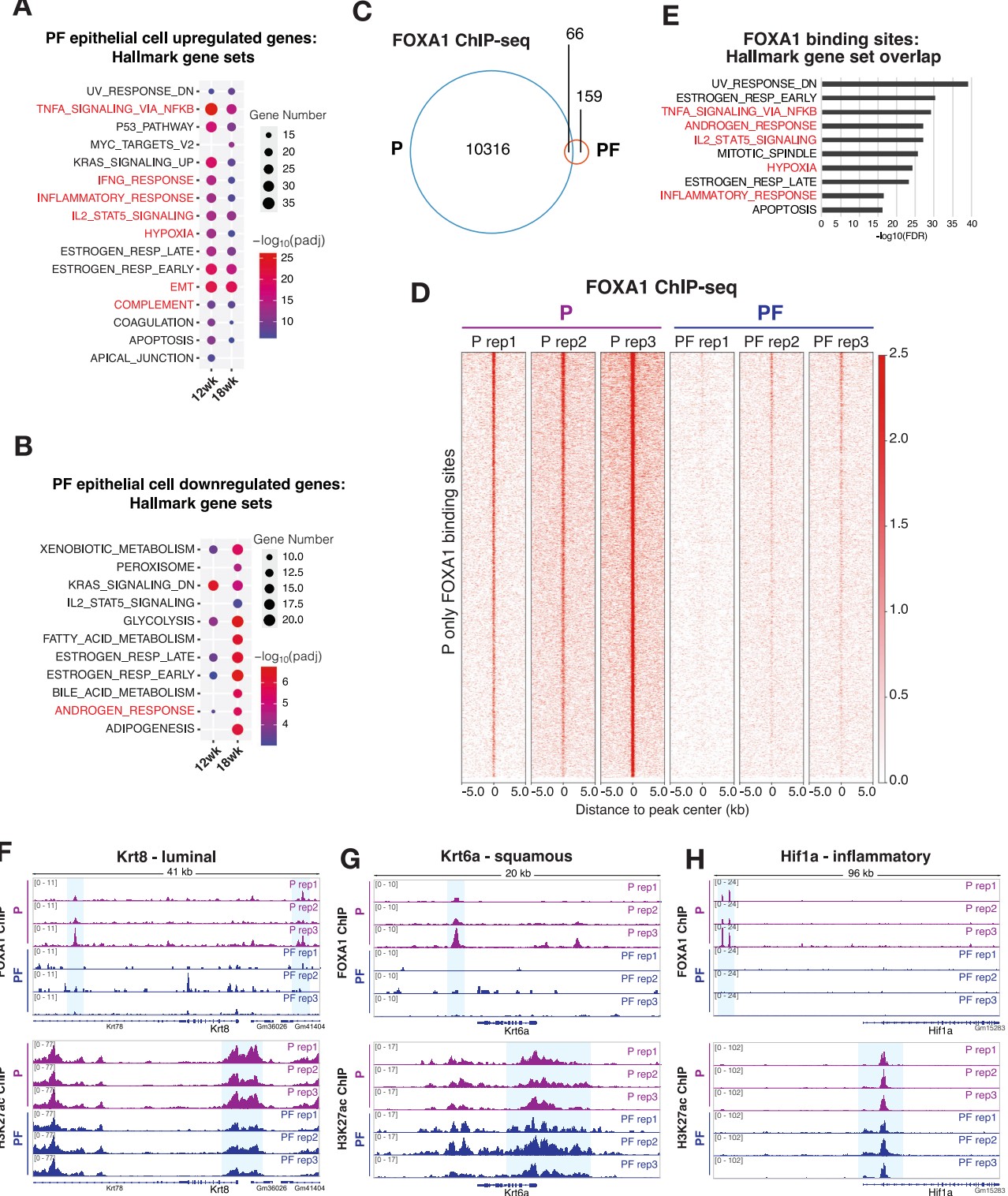

**Fig. 3 | FOXA1 binds the chromatin to directly induce luminal genes, but repress basal/squamous and inflammatory genes. A** Gene ontology (GO) analysis was performed on genes upregulated (log$_2$ fold change > 1) in PF (PbCre4:*Pten*$^{f/}$ $^f$*Foxa1*$^{f/f}$) vs. P (PbCre4:*Pten*$^{f/f}$) epithelial cells from the scRNA-seq dataset at 12wk (P12, $n$ = 1863 cells; PF12, $n$ = 2407 cells) and 18wk (P18, $n$ = 2252 cells; PF18, $n$ = 2445 cells) timepoints to assess enrichment of MSigDB Hallmark gene sets. Pathways associated with inflammatory cytokine signaling and EMT are indicated in red. Gene set overlap significance was assessed using a hypergeometric test (MSigDB overlap) with false discovery rate (FDR) correction. Gene Number indicates the number of genes in the overlap. **B** GO analysis of PF vs. P epithelial cells downregulated genes (log$_2$ fold change < −1), at 12wk and 18wk timepoints, to assess enrichment of MSigDB Hallmark gene sets. Known FOXA1-induced androgen response pathway is

indicated in red. Gene set overlap significance was assessed using a hypergeometric test with FDR correction. **C** Venn diagram of P only (10,316), PF only (159), and shared (66) FOXA1-binding sites identified from FOXA1 ChIP-seq performed in three independent biological replicates of age-matched 18wk P vs. PF mouse prostates. **D** Heatmaps showing FOXA1 ChIP-seq intensity across the three biological replicates of age-matched 18wk P and PF mouse prostates. **E** GO analysis to determine Hallmark gene set enrichment in FOXA1 binding sites. Pathways associated with androgen response and inflammatory signaling indicated in red. **F** Genome browser tracks of FOXA1 and H3K27ac ChIP-seq intensity around luminal gene *Krt8*. **G** Genome browser tracks of FOXA1 and H3K27ac ChIP-seq intensity around squamous gene *Krt6a*. **H** Genome browser tracks of FOXA1 and H3K27ac ChIP-seq intensity around inflammatory gene *Hif1a*.

---

*KRT6A* expression (Figure S3F). In accordance, evaluation of previously published FOXA1 ChIP-seq data in LNCaP control vs. *FOXA1* KD cells uncovered FOXA1 binding peaks near basal/squamous lineage genes such as *KRT14* and *KRT6A/C* (GSE55007)[6] (Figure S3G). A similar trend was observed for inflammatory gene *HIF1A* (Figure S3F, G), as previously published[14]. Taken together, these findings support our mouse data showing FOXA1 as a transcriptional repressor of squamous and inflammatory gene expression in human PCa.

### *Foxa1* loss remodels the tumor immune microenvironment, leading to tumor infiltration by immunosuppressive myeloid cells and dysfunctional T cells

Our scRNA-seq analyses have revealed that, in addition to basal/squamous cells, immune cells were also increased in the prostates of intact PF vs. P mice (Fig. 2D). Considering that castration itself has profound effects on tumor-immune infiltration and composition[43–45], we decided to focus our analyses on the TIME of intact P vs. PF mice. To delve deeper into how *Foxa1* loss affects the TIME, we isolated immune cells (CD45 + ) from the whole prostates of P15 and PF15 mice, a middle stage between 12wk and 18wk mice used for 10x scRNA-seq above, for PIP-seq scRNA-seq analysis. UMAP analyses and immune cell marker annotation identified several major immune populations, including Monocytes, Macrophages, Dendritic Cells (DC), B cells, CD8 T cells, CD4 T cells, and NK cells in the TIME of PCa (Fig. 4A, B). Of note, monocytes were poorly captured by PIP-seq as compared to 10x Genomics scRNA-seq, and there were more total immune cells captured from the PF relative to P mouse prostates (Figure S4A). Next, we focused on the CD8 and CD4 T cell populations and identified four CD8 T cell subclusters, including cycling CD8 T cells with high expression of cell cycle genes and three other cytotoxic T lymphocyte populations (CTL1, CTL2, and CTL3), and three major subclusters within the CD4 population, including *Foxp3*+ Tregs, *Ccr7*+ naïve/memory T helper, and *Rorc* + Th17 cells (Figure S4B, C). Interestingly, we found that PF prostates contained a greater proportion of CD4 T cells, as compared to CD8 T cells, a large percentage of which were of the immunosuppressive Treg or Th17 phenotype (Fig. 4C) that have been associated with impaired anti-tumor immune response in PCa[21–23]. Moreover, the CD8 T cell population in PF prostate tumors showed a decreased cytotoxic signature gene expression compared to P (Fig. 4D). On the other hand, we observed an increase in immunosuppressive M2 signature gene expression, along with a slight decrease in M1 anti-tumor signature gene expression, in tumor-associated macrophages (TAMs) of PF compared to P mice (Fig. 4E). In aggregates, these data suggest that *Foxa1* loss in PCa epithelial cells leads to T-cell dysfunction and promotes M2 polarization of TAMs.

To validate these results and further evaluate tumor immune infiltrates at protein levels, we performed multiparameter FACS staining and Cytek spectral flow analysis of whole prostates from several P ($n$ = 6) and PF ($n$ = 7) mice between the ages of 18–20 weeks (Figure S4D). We observed substantially increased monocytic myeloid-derived suppressor cell (M-MDSC) and polymorphonuclear MDSC

(PMN-MDSC) populations as a percentage of total CD45+ immune cells in PF compared to P tumors (Fig. 4F). Further, we noticed statistically significant increases in PD-L1 expression by M-MDSCs and PMN-MDSCs, along with DCs and TAMs, in PF tumors, suggesting that *Foxa1* loss drives myeloid populations to a more immunosuppressive state (Fig. 4G). Moreover, TAMs of PF tumors were found to exhibit higher expression of M2-like marker CD163 and lower expression of M1-like marker CD80, as compared to P tumors (Fig. 4H), further supporting that *Foxa1* loss promotes M2 polarization of TAMs. Next, we evaluated CD8 T cell phenotypes and found that PF tumors were more highly infiltrated by terminally exhausted T cells (TOX +/PD1+ and CD101 +/PD1+ subsets), while P tumors have more progenitor exhausted CD8 T cells (CD69 +/Ly108+ and PD1 +/Ly108+ subsets)[46], suggesting that *Foxa1* loss promotes terminal exhaustion of CD8 T cells within the PCa TIME (Fig. 4I). Accordingly, there was a decrease of TNFα +/IFNγ+ effector CD8 T cells upon *Foxa1* loss in their tumors, although not significant (Fig. 4J). Altogether, our data indicate immunosuppressive myeloid cell enrichment and a weakened CD8 T cell response or dysfunction in *Foxa1*-depleted PCa.

### Spatial Transcriptomics analyses revealed massive myeloid infiltration but constrained T-cell activities in *Foxa1*-depleted PCa

To understand the spatial organization of PCa and immune cells, we performed 10x Genomics Visium Spatial Transcriptomics (ST) analysis of prostate tissue sections derived from age-matched 12wk- and 18wk-old PF and P mice. First, ST data confirmed depleted *Foxa1* expression across the entire prostate section of the PF18 mouse (Figure S5A). We next utilized the TESLA machine-learning framework[47], which integrates spatial gene expression with tissue histology information to provide pixel-level resolution for annotating and characterizing different cell types within the TIME. We found that most cells in the P prostates were called luminal cells by the program, with minimal basal cells identified, as expected for prostate tissues (Fig. 5A). In line with our histopathology analyses, P prostate tumors appear to maintain their glandular architecture, with tumor epithelial cell regions surrounded by mesenchymal stromal cells (Fig. 5A & S5B, C). Critically, PF18 tumors exhibited a striking increase in basal-like cell density compared to P18 tumors, whereas luminal cells remained largely the same across genotypes, because many cells that were identified as basal cells in the PF tumors continued to express luminal genes and were simultaneously called luminal cells by TESLA, representing a mixed basal-luminal lineage. Interestingly, TESLA analysis for mesenchymal populations in PF tumors provided further evidence of lineage plasticity, with broad regions of tumor epithelial cells identified to express mesenchymal markers (Fig. 5A & S5C). Altogether, our ST data demonstrated that *Foxa1* loss leads to lineage plasticity and luminal-to-basal de-differentiation of PCa cells, being consistent with histology and scRNA-seq data shown earlier.

Next, we sought to evaluate the spatial distribution of immune cells in the TIME of PCa. Using TESLA to call immune cell densities across PF18

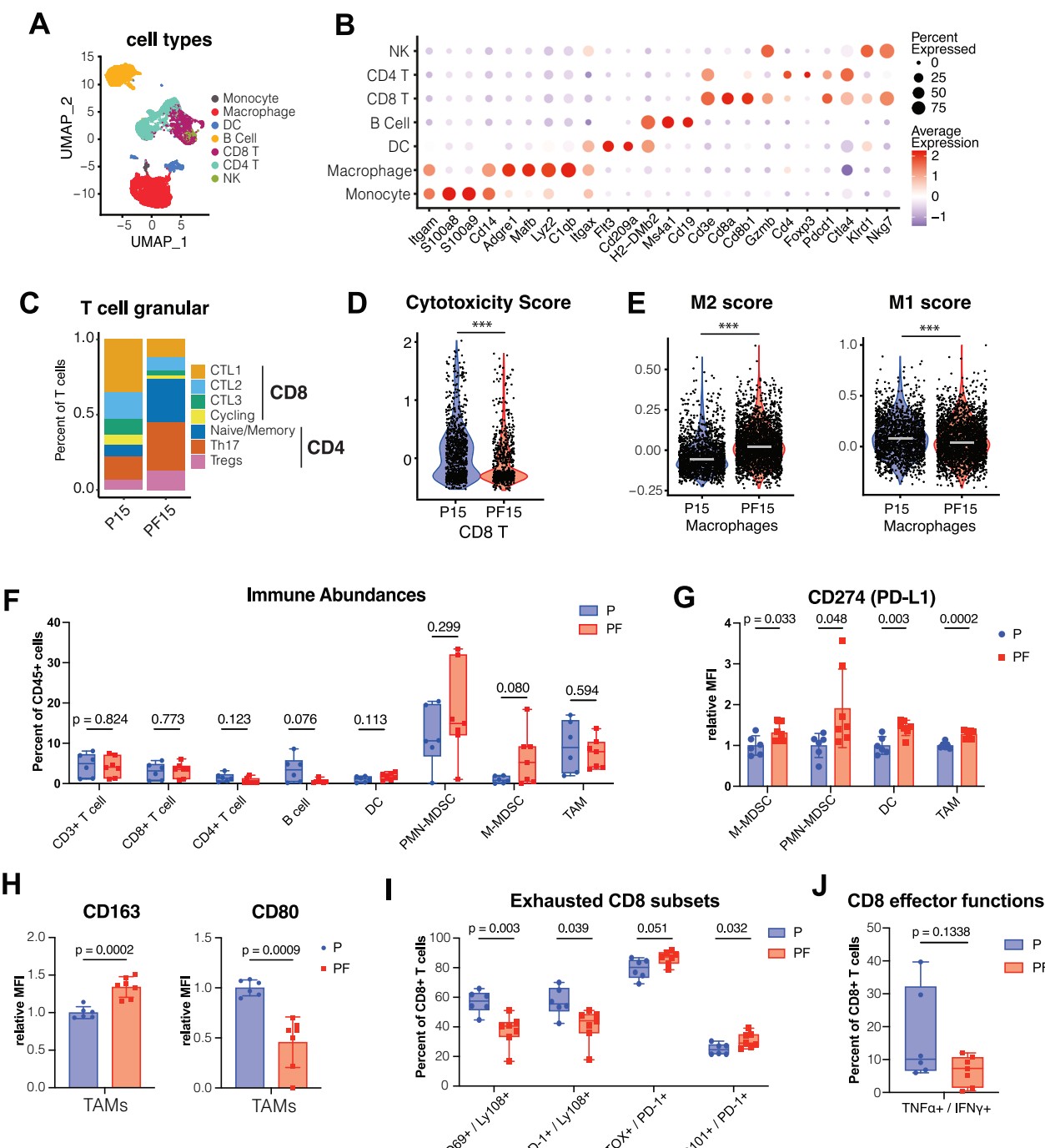

and P18 tissue sections, we identified striking increases in the accumulation of both myeloid and lymphoid cells upon *Foxa1* loss (Fig. 5B). Specifically, the abundance of TAMs, as a percentage of the total captured area, increased from 2.8% in P18 to 17.2% in PF18 PCa tissues. Similarly, monocytes increased from 12.8% in P18 to 26.1% in PF18 samples. We also observed a clear increase in T cell accumulation upon *Foxa1* depletion, from 2.4% CD4 + T cells and 0.9% CD8 + T cells in P18 to 19.5% and 19.7% in the PF tissue, respectively. However, we found that myeloid and T cell populations exhibited distinct distribution patterns in the TIME of PF mice. While T cells largely localized to stromal regions adjacent to tumor cell nests, macrophages and monocytes were more widely distributed throughout and into the tumor regions (Fig. 5C). Similar trends were also observed in the prostates of the 12-week-old mice (Figure S5D, E). Moreover, these findings were confirmed by IHC staining for F4/80+ macrophages, CD8 + T cells, and CD4 + T cells, which

again showed that T cell populations were largely restricted within intratumoral stromal regions whereas macrophages were well infiltrated into tumor cell nests (Fig. 5D). This suggests that, although *Foxa1* loss induced both macrophages and T cells, it promotes an "excluded" phenotype of T cells, which could have strongly limited T-cell anti-tumor activities. Finally, further phenotyping of the major immune populations identified in our ST analysis confirmed that PF immune infiltrates exhibit an immunosuppressive phenotype. For instance, the majority of macrophages in PF tumors were found to express M2 markers (Fig. 5E & S5F), and a larger proportion of CD8 T cells expressed exhaustion markers compared to cytotoxic markers (Fig. 5F & S5G). Altogether, ST data confirmed luminal-to-basal de-differentiation of PCa cells and showed an immunosuppressive TIME caused by massive myeloid infiltration in the tumors but spatially constrained T-cell activities in the *Foxa1*-depleted PCa.

**Fig. 4 | *Foxa1* loss promotes immunosuppressive myeloid cell accumulation and T cell dysfunction. A** UMAP of CD45+ sorted cells from a pair of P15 (15wk PbCre4:*Pten^{f/f}*, *n* = 3564 cells) and PF15 (15wk PbCre4:*Pten^{f/f}Foxa1^{f/f}*, *n* = 8991 cells) PIP-seq scRNA-seq samples with cell type annotations. **B** Expression of key immune cell type markers within identified cell type clusters. **C** Proportion of CD4 and CD8 T cell subclusters in P15 vs. PF15 tumors. **D** Cytotoxicity score of CD8 T cells populations in P15 vs. PF15 samples, shown as violin plots. Statistical significance (*p* = 9.79e-21) determined using ggbetweenstats, two-sided Mann-Whitney nonparametric test. P15, *n* = 947 CD8 T cells; PF15, *n* = 797 CD8 T cells. (***p*-value < 0.001, ***p*-value < 0.01, **p*-value < 0.05.). **E** Violin plot depicting M2 and M1 signature gene expression in macrophages from P15 vs. PF15 tumors. Statistical significance (M1: *p* = 9.12e-08; M2: *p* = 1.52e-97) determined using ggbetweenstats, two-sided Mann-Whitney nonparametric test. P15, *n* = 1794 macrophage cells; PF15, *n* = 2628 macrophage cells. (***p*-value < 0.001, ***p*-value < 0.01, **p*-value < 0.05). **F** Cytek spectral flow analysis of immune cell abundance in P (*n* = 6 mice) vs. PF (*n* = 7 mice) prostate tumors, 18–21wks of age, shown as percent of total CD45+ cells. Statistical significance was determined by multiple unpaired t-tests with Welch correction. Box plots show the median (center line), interquartile range (box; 25th–75th percentiles), and whiskers representing minimum and maximum values, with individual points shown. **G** Cytek spectral flow analysis of immunosuppressive CD274 (PD-L1) expression in M-MDSCs (monocytic myeloid-derived suppressor cells), PMN-MDSCs (polymorphonuclear myeloid-derived suppressor cells), DCs (dendritic cells), and TAMs (tumor-associated macrophages) in PF vs. P tumors. Relative mean fluorescence intensity (MFI) for CD274 is depicted in bar plots, where samples were normalized to the MFI of the P condition. Data are depicted as mean + SD error bars with individual points shown. Statistical comparisons between groups were performed using two-sided unpaired t-tests with Welch correction. **H** Cytek spectral flow analysis of M2 marker CD163 and M1 marker CD80 expression in PF vs. P TAMs. Relative MFI is depicted in bar plots, where samples were normalized to the MFI of the P condition. Data are depicted as mean + SD error bars with individual points shown. Statistical comparisons between groups were performed using two-sided unpaired t-tests with Welch correction. **I** Cytek spectral flow analysis of progenitor exhausted CD69 + /Ly108+ and PD1 + /Ly108 + CD8 T cells and terminally exhausted TOX + /PD1+ and CD101 + /PD1 + CD8 T cells in PF vs. P, as a percentage of total CD8 + T cells. Statistical comparisons between groups were performed using two-sided unpaired t-tests with Welch correction. Box plots show the median (centre line), interquartile range (box; 25th–75th percentiles), and whiskers representing minimum and maximum values, with individual points shown. **J** Cytek spectral flow analysis of TNFα + /IFNγ+ effector T cells as a percentage of CD8 + T cells in PF vs P tumors. Statistical comparisons between groups were performed using two-sided unpaired t-tests with Welch correction. Box plots show the median (centre line), interquartile range (box; 25th–75th percentiles), and whiskers representing minimum and maximum values, with individual points shown. (Source data for 4F-J are provided in the Source Data file.)

## Foxa1 loss greatly enhanced immunosuppressive cytokine signaling, primarily from basal cells to macrophages

We next sought to understand how *Foxa1* deletion within epithelial cells leads to the immunosuppressive phenotype observed in PF tumors. To study the communication between cancer cells and immune populations in the TIME, we utilized the bioinformatics tool CellChat[48,49], which estimates ligand-receptor interactions between cell groups from scRNA-seq data based on gene expression and literature on ligand-receptor pairs. We focused our search on the "Secreted Signaling" subset of CellChat's ligand-receptor database, considering the important roles of cytokine signaling in regulating tumor immunity. First, comparing the number of secreted signaling interactions among epithelial, immune, and stromal cells between PF and P prostates, we observed a major increase in epithelial-to-immune/stroma interactions upon *Foxa1* loss at both 12-wk and 18-wk timepoints (Fig. 6A). Further, by investigating individual secreted signaling pathways in PF vs. P tumors, we found that a vast majority of these pathways have increased interactions among cell types in PF tumors, some of which, such as PTN, KIT, and IL6/1, are exclusive to PF tumors, indicating that they only mediate interactions between cell groups in PF tumors (Fig. 6B & S6A). To focus on the pathways through which FOXA1 might regulate how epithelial cells interact with immune cells, we cross-referenced the PF-enriched pathways with those originating from luminal and basal-like populations and identified three major pathways of interest: TGF-β, PTN, and MIF (Fig. 6C & S6B). Notably, signaling through these pathways appears to be more strongly contributed by the basal-like populations than luminal, suggesting a link between *Foxa1*-loss-induced basal phenotype and increased secreted signaling.

To further decipher how epithelial and/or immune populations may be signaling with each other through these pathways in *Foxa1*-deficient tumors, we determined their signaling networks among major cell types, including luminal, basal, T cell, B cell, macrophage, monocyte, and stromal populations in the PF mice. Critically, we observed dominant interactions originating from basal cells to other cell types for all 3 pathways, with PTN signaling originating solely from basal cells and sending signals to all other cell types, including macrophages and monocytes, and MIF signaling originating from basal/luminal epithelial cells to mostly macrophages (Figure S6C), being consistent with their reported roles in recruiting tumor-promoting neutrophils and macrophages, respectively[50,51]. Importantly, TGF-β signaling stands out as a major pathway through which *Foxa1*-depleted prostate epithelial cells, especially the basal-like population, sent and also received signals from immune and stromal cell populations in the TIME, with dominant signaling from basal to luminal, stromal, and macrophage cells (Fig. 6D). Accordingly, TGF-β ligands, including TGF-β1, 2, 3, are well-known immunosuppressive cytokines that prompt M2 polarization, Treg induction, and effector T cell inhibition[52–55] and induce reactive stroma in PCa[40].

Next, we evaluated the expression of TGF-β cytokines in our scRNA-seq data and found that *Foxa1* loss drastically induced the expression of *Tgfb3*, *Tgfb2*, and *Tgfb1* in luminal and basal-like cells of the prostate tumors (Fig. 6E, F & S6D). *Tgfb2* and *Tgfb3* were mainly expressed in epithelial cells and stroma, while *Tgfb1* was most highly expressed in immune cells, such as macrophages and monocytes, supporting TGF-β signaling originated from these various cell types in CellChat analyses. As TGF-β cytokines signal through their receptors, we also examined receptor expression and confirmed that both *Tgfbr1/2* were expressed in almost all cells of P and PF tumors, with immune cells and stroma having the highest expression (Figure S6E), in agreement with their primary roles as receivers in cell-cell communication. To further confirm this finding that TGF-β cytokine genes are induced upon *Foxa1* loss, we evaluated the ST analysis of P18 vs. PF18 tumor sections and observed a significant upregulation of *Tgfb3*, *Tgfb2*, and *Tgfb1* cytokine genes in PF18 compared to P18 tumors (Figure S6F, G). Moreover, ChIP-seq data revealed some FOXA1 binding events at or near TGF-β cytokine genes in P tumors, which were diminished in the PF tumors (Figure S6H), suggesting that FOXA1 directly represses TGF-β cytokine gene expression. Finally, to confirm this phenomenon at the protein level, we stained P18 vs. PF18 tumor sections for pSMAD2, a downstream readout for TGF-β signal transduction. As expected, pSMAD2 staining was drastically increased in PF18 tumors, as compared to P18 (Fig. 6G). Further, staining for F4/80+ macrophages demonstrated increased macrophage accumulation in pSMAD2-high PF18 tumors, and reduced macrophage accumulation pSMAD2-low P18 tumors (Fig. 6G), further supporting increased TGF-β signaling and macrophage accumulation upon *Foxa1* loss. Taken together, our data revealed drastically increased cell-cell communication in PF tumors, in particular, immunosuppressive cytokine signaling from basal cells to macrophages, suggesting that *Foxa1* loss leads to an immunosuppressive TIME through increased cytokine signaling that is likely caused by luminal-to-basal cell de-differentiation.

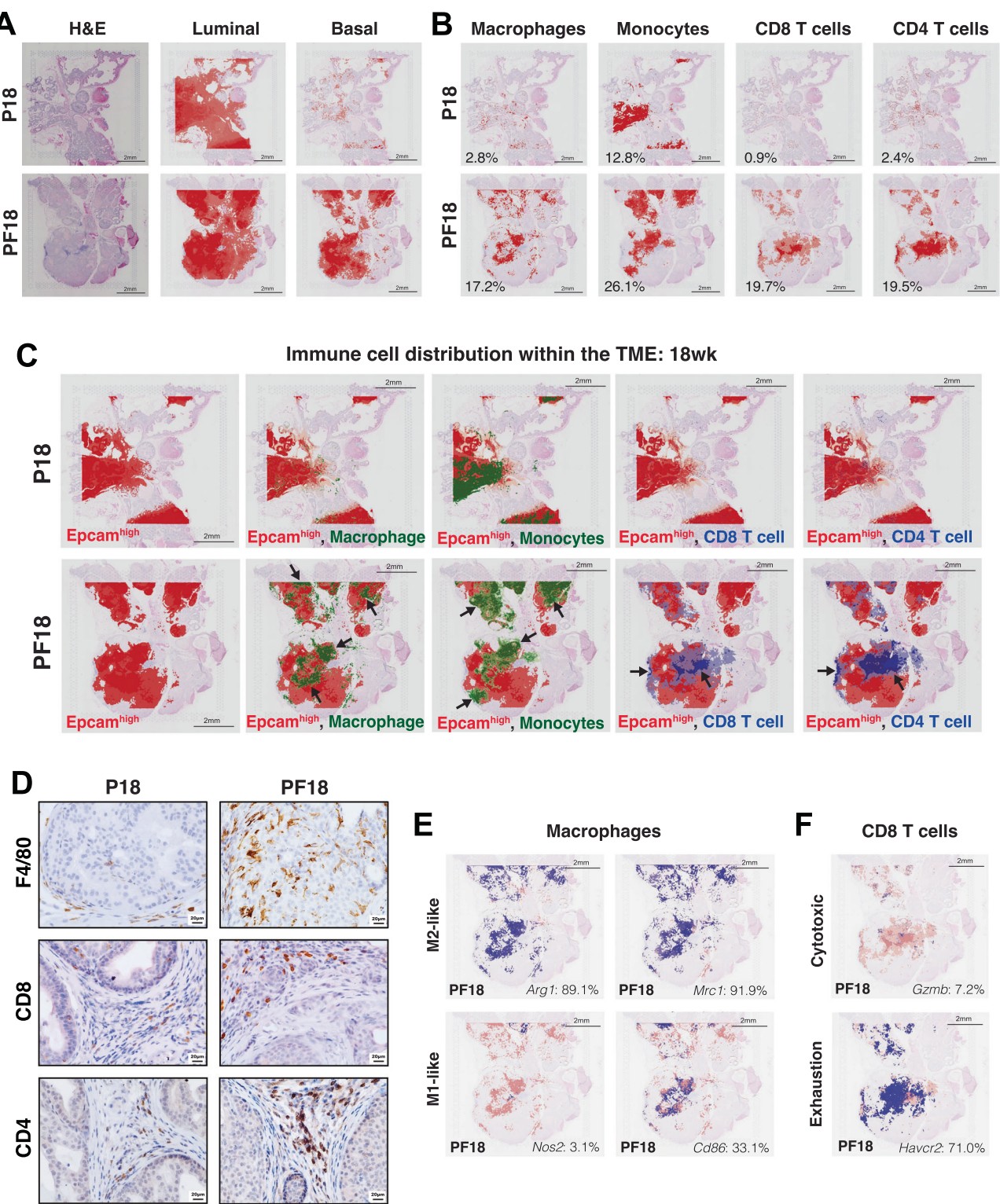

**C** Immune cell distribution within the TME: 18wk

To directly investigate whether FOXA1 loss drives macrophage infiltration through a TGF-β-mediated pathway, we performed an in vitro co-culture assay using a prostate cancer cell line (PC3) that was transfected with either an empty vector (PC3 – WT) or *FOXA1* knock-down vector (PC3 – *FOXA1* KD). We utilized the Stacks cell culture platform, which is an open, reconfigurable microfluidic cell culture platform that has been validated for the investigation of the TIME[56–58]. PCa cells were cultured with primary monocyte-derived macrophages (MDMs), derived from peripheral blood samples of patients with PCa,

in the presence or absence of the TGF-β receptor I inhibitor (TGFBi), Galunisertib. After 24 h, the MDMs were analyzed for tumor-directed migration, and PC3 – WT and PC3 – *FOXA1* KD cells were analyzed for mRNA expression.

Analysis of the mRNA expression within the tumor cells confirmed downregulation of *FOXA1* mRNA expression and supported the upregulation of *CCL2*, *MIF*, and *TGFB3* in tumor cells with *FOXA1* loss (Figure S6I), in line with our mouse model scRNA-seq analyses and previous findings[9,14]. Moreover, analysis of MDM migration using in-

**Fig. 5 | Spatial transcriptomics analyses revealed massive myeloid infiltration but constrained T cell activities in *Foxa1*-depleted PCa. A** Spatial transcriptomics analysis by 10x Genomics Visium was performed on P18 (18wk PbCre4:*Pten^f/f*) and PF18 (18wk PbCre4:*Pten^f/f*Foxa1^f/f*) samples. H&E staining is shown, along with Luminal and Basal cells determined using TESLA. A darker red indicates a greater relative abundance. **B** TESLA annotation of macrophages (*Adgre1/Mafb/Lyz2/C1qb*), monocytes (*Cd14/S100a8/S100a9*), CD4 T cells (*Cd4*), and CD8 T cells (*Cd8a/Cd8b1*) in P18 vs. PF18 tissue sections profiled by Visium. Percentages depicted represent the percent of total captured area classified as enriched for the indicated cell type. **C** Epithelial tumor regions (*Epcam^high*) depicted in red were overlaid with macrophages (green), monocytes (green), CD4 T cells (blue), or CD8 T cells (blue) to visualize immune cell distribution relative to PCa cells in P18 vs. PF18. **D** IHC staining for F4/80+ macrophages, CD8 + T cells, and CD4 + T cells in P18 vs PF18 tissue sections (40X). Similar staining patterns were observed in at least 3 independent biological replicates per genotype. **E** TESLA analysis of the percent of the total macrophage population (light red) in the PF18 sample enriched for M2-like gene expression (*Arg1* + 89.1%, *Mrc1* + 91.9%, blue, top panel) as compared to M1-like genes (*Nos2* + 3.1%, *Cd86* + 33.1%, blue, bottom panel). **F** TESLA analysis of the percent of the total CD8 T cell population (light red) identified in the PF18 sample enriched for cytotoxic marker (*Gzmb* + 7.2%, blue, top panel) or exhaustion marker genes (*Havcr2* + 71.0%, blue, bottom panel). **A**–**C**, **E**, **F** One mouse prostate tumor per genotype was analyzed by Visium at each timepoint (12 and 18 weeks; P12, PF12, P18, PF18), with similar trends observed across timepoints.

chip, fluorescence confocal microscopy demonstrated a significant increase in the number of MDMs that migrated when cultured with the PC3 – *FOXA1* KD PCa cells compared with the PC3 – WT cells. Furthermore, the increase in MDM migration in the PC3 – *FOXA1* KD co-culture condition was abrogated in the wells that were treated with TGFBi (Fig. 6H). These findings provide direct evidence that loss of FOXA1 in tumor cells drives tumor-directed migration of primary, PCa-associated MDMs through a TGF-β-mediated process.

Finally, to evaluate whether targeting FOXA1-loss induced TGF-β signaling reduces tumor growth in vivo, we treated mice harboring subcutaneous PF tumor-derived allografts with TGFBi (Galunisertib) or a vehicle control for three weeks. Notably, TGFBi treatment led to a reduction in tumor weight compared to vehicle control, though not significant, suggesting TGFBi treatment inhibits *Foxa1*-deficient PCa tumor growth (Figure S6J). Taken together, these results support that *Foxa1* loss drives an immunosuppressive TIME and PCa progression by enhancing TGF-β signaling.

### *FOXA1* downregulation in human PCa is associated with epithelial lineage switch and immunosuppression

To assess the clinical relevance of our mouse model findings in PCa patients, we examined gene expression in 179 clinical primary PCa samples[59] and found that *FOXA1* expression is indeed negatively correlated with the expression of immune markers, such as *CD68*, *CD8A*, *CD4*, and *FOXP3*, as well as TGF-β cytokine genes (Fig. 7A). To further probe FOXA1's relation to immune infiltration in CRPC, we utilized the CIBERSORTx[60] deconvolution tool to analyze the Stand-up-to-cancer (SU2C) CRPC dataset[61] to estimate the abundance of immune cell types in each of the 208 CRPC samples. Interestingly, our results revealed that *FOXA1* expression was significantly negatively correlated with the abundance of various immune cell populations, supporting an overall trend towards increased immune cell accumulation in *FOXA1*-low tumors (Fig. 7B). Notably, M2 macrophage was the immune cell type that is the most significantly negatively correlated with *FOXA1*, in line with our mouse data that showed increased M2-polarization of TAMs in *Foxa1*-depleted tumors. This bulk RNA-seq data also showed an increase of T cells, but mostly Tregs and naïve/memory CD4 T cells. Moreover, T cells might also be spatially restricted in the stroma surrounding the tumor regions, further limiting their anti-tumor activities, as we have found in mouse models.

Next, to determine how *FOXA1* expression in epithelial cells relates to lineage-specific gene expression and cytokine signaling in human PCa, we analyzed scRNA-seq data of 17 human localized PCa tumor samples (GSE181294)[21]. We focused on gene expression within the epithelial cell populations and stratified the 17 PCa samples into *FOXA1*-high (*n* = 8) and *FOXA1*-low (*n* = 9) groups based on epithelial *FOXA1* expression in each sample (Figure S7A). We found that androgen signaling-related genes such as *AR* and *KLK3* were downregulated, while basal/squamous genes *TP63*, *KRT5*, and *KRT13* were upregulated in the epithelial cells of *FOXA1*-low PCa (Fig. 7C). Overall, pathways related to lineage plasticity or squamous differentiation, for instance, EMT and Skin Development gene sets, were upregulated in the *FOXA1*-low group, while androgen response pathways were downregulated (Fig. 7D), supporting our findings in mouse models that *Foxa1* loss induces basal/squamous de-differentiation. Moreover, TGF-β signaling, Inflammatory response, TNFα signaling, and Hypoxia Hallmark gene sets were more highly expressed by epithelial populations in the *FOXA1*-low group (Fig. 7E & S7B), validating our observations in mice that *Foxa1* loss increases immunosuppressive cytokine signaling. Altogether, our patient data analysis supports our observations of epithelial lineage switch, increased inflammatory and cytokine signaling, tumor immune cell accumulation, and immunosuppression upon FOXA1 loss.

## Discussion

The literature has shown controversial roles of FOXA1 and its mutations in inhibiting or promoting PCa progression[2,3,10,13]. To address this fundamental question for a master regulator of PCa, we generated a genetically engineered mouse model where we deleted *Foxa1* in *Pten*-null mice. As FOXA1 is initially up-regulated in primary PCa and then downregulated as the disease progresses to CRPC[6,8–10], we expected PF mice to exhibit reduced tumor progression in the intact mice, but greatly increased aggressiveness upon castration. Surprisingly, however, our results demonstrated a much more aggressive PCa phenotype and reduced survival in the PF mice, even under intact condition, compared to P mice. There could be multiple reasons for this more tumorigenic role of *Foxa1* loss in the PF mice than in previous studies mostly based on cell culture, organoids, or subcutaneous xenograft models[2,3,10,62]. First, previous studies have reported that intact mice already exhibit quite low endogenous androgen levels, comparable to those seen in human CRPC patients who have undergone androgen deprivation therapy[38]. Second, in addition to a reduction of androgen response as previously reported in vitro[5], there is an apparent luminal to basal/squamous de-differentiation in the *Foxa1*-depleted PCa tumors, which is consistent with FOXA1's established role as a luminal transcription factor but has not been reported in previous PCa studies. Squamous differentiation, though rare, has been observed in treatment-resistant PCa and is associated with poor prognosis[63,64]. Likewise, squamous differentiation is often observed in *EGFR* and *KRAS* mutant lung cancers becoming resistant to their respective EGFR or KRAS targeted therapies[65,66]. Lastly, the more aggressive phenotype observed in the intact PF mice vs. previous studies could be due to the use of an immune-competent autochthonous PCa mouse model in this study, wherein *Foxa1* depletion within a prostate-specific tumor context leads to an immunosuppressive TIME, promoting tumor growth. Therefore, by here generating a genetically engineered mouse model of PCa, we establish FOXA1 as a tumor suppressor gene, critical for maintaining proper epithelial lineage and inhibiting aggressive PCa.

We notably observed a striking reprogramming of tumor epithelial cells from a luminal-like to a more inflammatory basal/squamous-like phenotype upon *Foxa1* loss. Interestingly, a recent study using mouse prostate-derived organoids found that retinoic acid (RA) signaling through RARγ enforces luminal identity in epithelial progenitor cells by promoting FOXA1 expression. In line with our findings, this

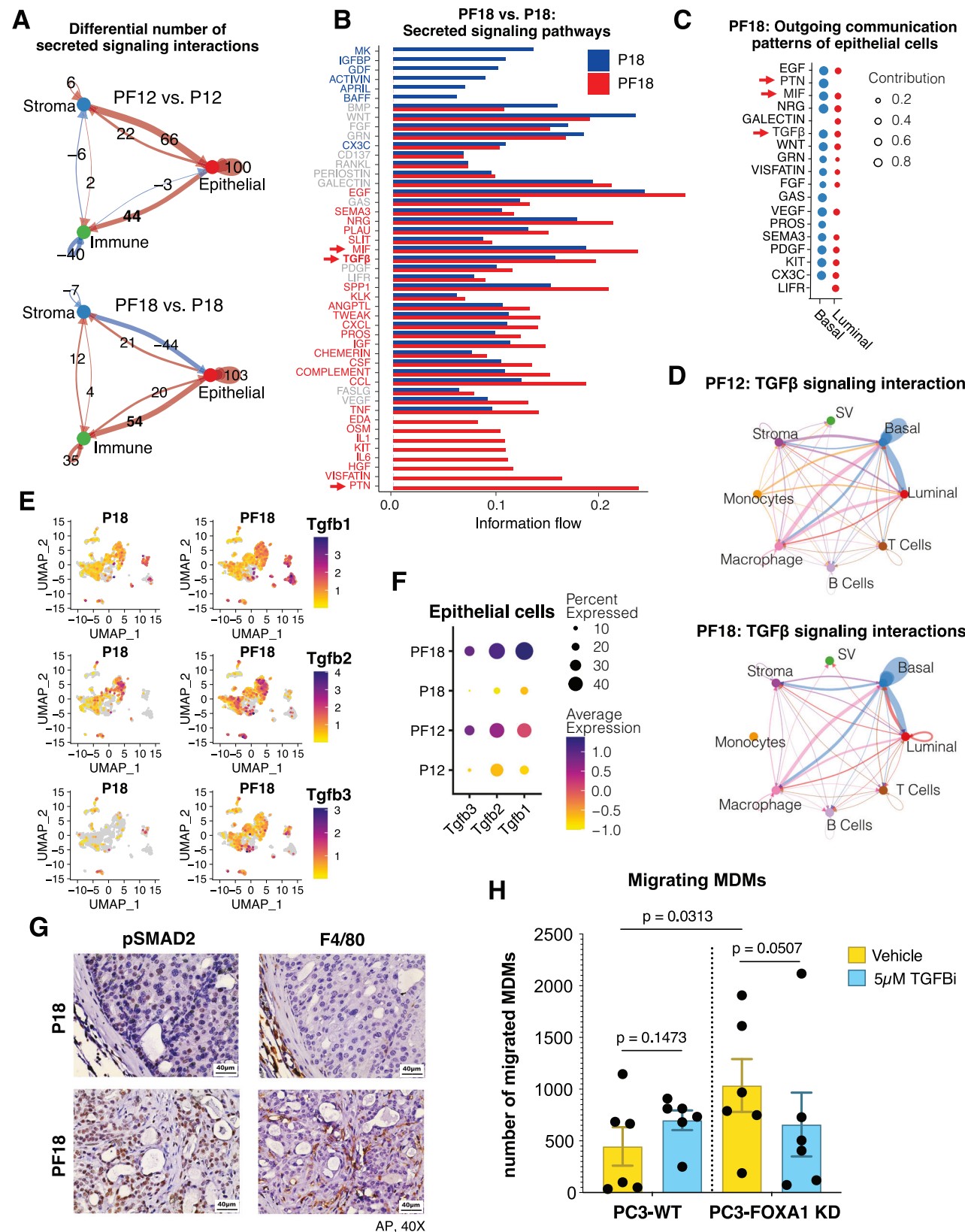

**A** Differential number of secreted signaling interactions

**B** PF18 vs. P18: Secreted signaling pathways

**C** PF18: Outgoing communication patterns of epithelial cells

**D** PF12: TGFβ signaling interactions

PF18: TGFβ signaling interactions

**E** P18 · PF18 · Tgfb1 · Tgfb2 · Tgfb3

**F** Epithelial cells

**G** pSMAD2 · F4/80 · AP, 40X

**H** Migrating MDMs

study observed that disrupting RA-RARγ-FOXA1 signaling axis via a cancer-associated FOXA1 loss-of-function mutation led to prostate progenitor cell de-differentiation and lineage plasticity[67]. Moreover, a recent study in bladder cancer identified a similar association between FOXA1 loss and squamous differentiation[68], suggesting that FOXA1's role in repressing squamous differentiation is not limited to PCa. Other

studies in bladder cancer have also identified a more inflamed and aggressive phenotype in basal/squamous as compared to luminal tumors[69], similar to our observations in the PF mice. Moreover, several recent studies have linked tumor lineage plasticity to inflammatory signaling and cellular crosstalk within the TIME. For example, Chan et al. reported that PCa lineage plasticity is dependent on

**Fig. 6 | *Foxa1* loss enhances basal-to-immune cell immunosuppressive cytokine signaling. A** CellChat analysis of the differential number of "secreted signaling" interactions among epithelial, immune, and stromal populations in PF12 (12wk PbCre4:*Pten*^f/f^*Foxa1*^f/f^, *n* = 3169 cells) vs. P12 (12wk PbCre4:*Pten*^f/f^, *n* = 2263 cells) and PF18 (18wk PbCre4:*Pten*^f/f^*Foxa1*^f/f^, *n* = 3584 cells) vs. P18 (18wk PbCre4:*Pten*^f/f^, *n* = 2738 cells) scRNA-seq data. Arrows point from the origin cells to the receiver cells. Red indicates interactions enhanced in the PF condition, and blue indicates interactions attenuated in the PF condition. **B** Overall information flow, defined by the sum of communications among all pairs of cell groups, for secreted signaling pathways (denoted on the left) ranked by their enrichment in P18 (blue) vs. PF18 (red) samples. **C** Dot plot showing major outgoing communication pathways of luminal-like and basal-like epithelial populations in the PF18 condition. Pathways identified as enriched in the PF condition and a major outgoing communication pattern of epithelial cells, at both 12wk and 18wk timepoints, are indicated with red arrows. **D** Circle plot depicting inferred source-target interactions within the TGF-β signaling pathway among cell type groups for PF18 and PF12 tumors. Arrow thickness signifies the interaction strength, and its color corresponds to its cell type source (blue originated from basal cells and pink from macrophages). **E** *Tgfb3*, *Tgfb2*, and *Tgfb1* gene expression projected onto integrated UMAP for P18 and PF18 samples. **F** Dotplot quantifying *Tgfb3*, *Tgfb2*, and *Tgfb1* average expression within epithelial cells and the percentage of epithelial cells expressing *Tgfb3*, *Tgfb2*, *and Tgfb1* in P12, PF12, P18, and PF18. **G** Representative IHC staining of pSMAD2 and F4/80 in P18 and PF18 tumor sections (AP lobe, 40X). Similar results were observed in 3 independent biological replicates per genotype. **H** Quantification of migrated patient-derived monocyte-derived macrophages (MDMs) in response to co-culture with PC3-WT or *FOXA1* knockdown (KD) cells, treated with vehicle or 5 μM of TGF-β receptor 1 inhibitor (TGFBi) LY2157299. *n* = 6 biological replicates (patient samples). Data presented as the mean ± SEM, with individual points shown. Statistical significance was evaluated using the Friedman test. Source data are provided in the Source Data file.

inflammatory signaling and found that targeting inflammatory IL6/JAK/STAT signaling restored luminal identity[70]. Another recent study in pancreatic cancer observed that inflammation induces epigenetic plasticity and identified an epithelial-immune cell interaction IL33 feedback loop driving *Kras*-mutant tumors[71]. We here observed that *Foxa1* loss markedly upregulates basal/squamous and inflammatory genes, and, therefore, we believe the luminal-to-basal/squamous lineage switch and the gain of inflammation together lead to a much more aggressive phenotype in the PF mice.

The potential roles of FOXA1 in regulating the TIME are understudied and have been largely limited by the lack of autochthonous PCa mouse models. Previous studies have been limited to in vitro and/or subcutaneous syngeneic mouse models, where FOXA1 loss has been shown to recruit macrophages under hypoxia conditions and increase T cell activation upon treatment using IFN inducers, together suggesting its regulation of both macrophage and T cells[14,15]. Utilizing the PF genetically engineered mice, here we found that *Foxa1*-depleted PCa induces immunosuppressive MDSCs, M2-like TAMs, and dysfunctional or exhausted T cells. We further showed that this immune modulation is largely associated with enhanced signaling of tumor-promoting cytokines, such as PTN, MIF, and TGF-β, largely from basal-like cells to macrophages. Moreover, powered by ST analysis, we observed PCa tumors to be largely infiltrated by TAMs, while T cells appear more excluded to the stroma surrounding the tumor cell nests. Such T cell exclusion is associated with poorer response to immunotherapies[54,72], highlighting the importance of elucidating the spatial organization of immune cells within the TIME. Squamous de-differentiation and the overall immunosuppressive TIME both support intrinsic resistance to immunotherapy, as found in bladder cancer[68]. However, the exhausted and spatially confined T cells might also be exploited for activation utilizing exogenic stimulus, such as IFN inducers, at least in model systems, to increase their tumor infiltration, and thus tumor response to immunotherapy, as previously reported[15]. On the other hand, the immunosuppressive cytokines might be targeted to inhibit macrophages and induce T cells to increase tumor response to ICIs[22,73]. Indeed, targeting TGF-β signaling led to reduced macrophage infiltration in vitro and reduced allograft tumor growth in vivo, supporting further investigation of TGF-β signaling as a target in FOXA1-low PCa.

Altogether, our study establishes FOXA1 as a tumor suppressor of PCa through its tumor cell-intrinsic and -extrinsic roles in regulating prostate epithelial cell lineage identity and the TIME and provides insights into the critical genes and pathways involved in these functions. There are some limitations to our study, including limited sample numbers for spatial transcriptomics, the sparse immune cells relative to tumor cells that were captured by 10x Genomics single-cell analyses of whole prostates, and the poor capture of certain immune cell populations by scRNA-seq platforms, which we addressed using Cytek Spectral Flow analyses. In future studies, it will also be interesting to target some of the cytokines, such as TGF-β, to see its effect on ICI response and examine how FOXA1 mutations affect PCa progression in vivo.

## Methods

### Ethics Statement

In vitro co-culture studies utilizing patient-derived monocytes were conducted in compliance with the Declaration of Helsinki. Patients were enrolled under institutional IRB-approved biospecimen protocols (1202-1214 and 2020-0915). Written informed consent was obtained from all participants before enrollment. The institutional biospecimen collection protocol does not allow unrestricted public access to the raw data to maintain protection of patient privacy. Therefore, data sharing requests must be submitted to the William S Middleton Memorial Veterans Hospital for review and approval.

All mouse work was approved by the Institutional Animal Care and Use Committee (IACUC) at Northwestern University and Emory University in compliance with all relevant ethical regulations.

### Generating genetically engineered mouse models

The PbCre4 transgene was utilized in this model, which enables expression of Cre recombinase under control of the probasin promoter, to allow Cre-Lox mediated deletion of target genes in prostate epithelial cells[36]. The PbCre4 mouse was first crossed with the *Foxa1*^f/f^ mouse, generously shared by Dr. David Degraff (Penn State Hershey Medical Center), to generate PbCre4:*Foxa1*^f/f^ mouse with prostate specific *Foxa1* KO[37]. Next, we crossed this mouse with the *Pten*^f/f^ mouse to generate the PbCre4:*Pten*^f/f^*Foxa1*^f/f^ mouse. And simultaneously crossed PbCre4 mouse with the *Pten*^f/f^ mouse to generate PbCre4:*Pten*^f/f^ mouse. The PbCre4 gene was only transmitted through male breeder mice. Only PbCre4 negative females were used for breeding, as maternally inherited PBCre4 can lead to nonspecific recombination of loxP-flanked alleles in other tissues[74]. Only heterozygous *Pten* KO male mice were used for breeding due to poor fertility of *Pten* homozygous KO male mice. Mice are of C57BL/6 background. Humane endpoints were defined by clinical criteria (e.g., weight loss, body condition, mobility, or distress), as tumors were not externally measurable. No animals exceeded approved tumor burden limits. Mice were maintained under standard housing conditions, including a 12-hour light/12-hour dark cycle, ambient temperature maintained at -19–23 °C, and relative humidity of 30–70%, with ad libitum access to food and water. The primers used for genotyping and the expected band sizes for wild-type and mutant alleles are as follows: Pten (F, CAA GCA CTC TGC GAA CTG AG; R, AAG TTT TTG AAG GCA AGA TGC; mutant=328 bp; heterozygote= 156 bp and 328 bp; WT = 156 bp), Foxa1 (F, CTG TGG ATT ATG TTC CTG ATC; R, GTG TCA GGA TGC CTA TCT GGT; mutant=480 bp; heterozygote= 290 bp and 480 bp; WT = 290 bp), and PBCre4 (F, CTG AAG AAT GGG ACA GGC ATT G; R, CAT CAC TCG TTG CAT CGA CC; Internal positive control F, CAA ATG TTG CTT GTC TGG

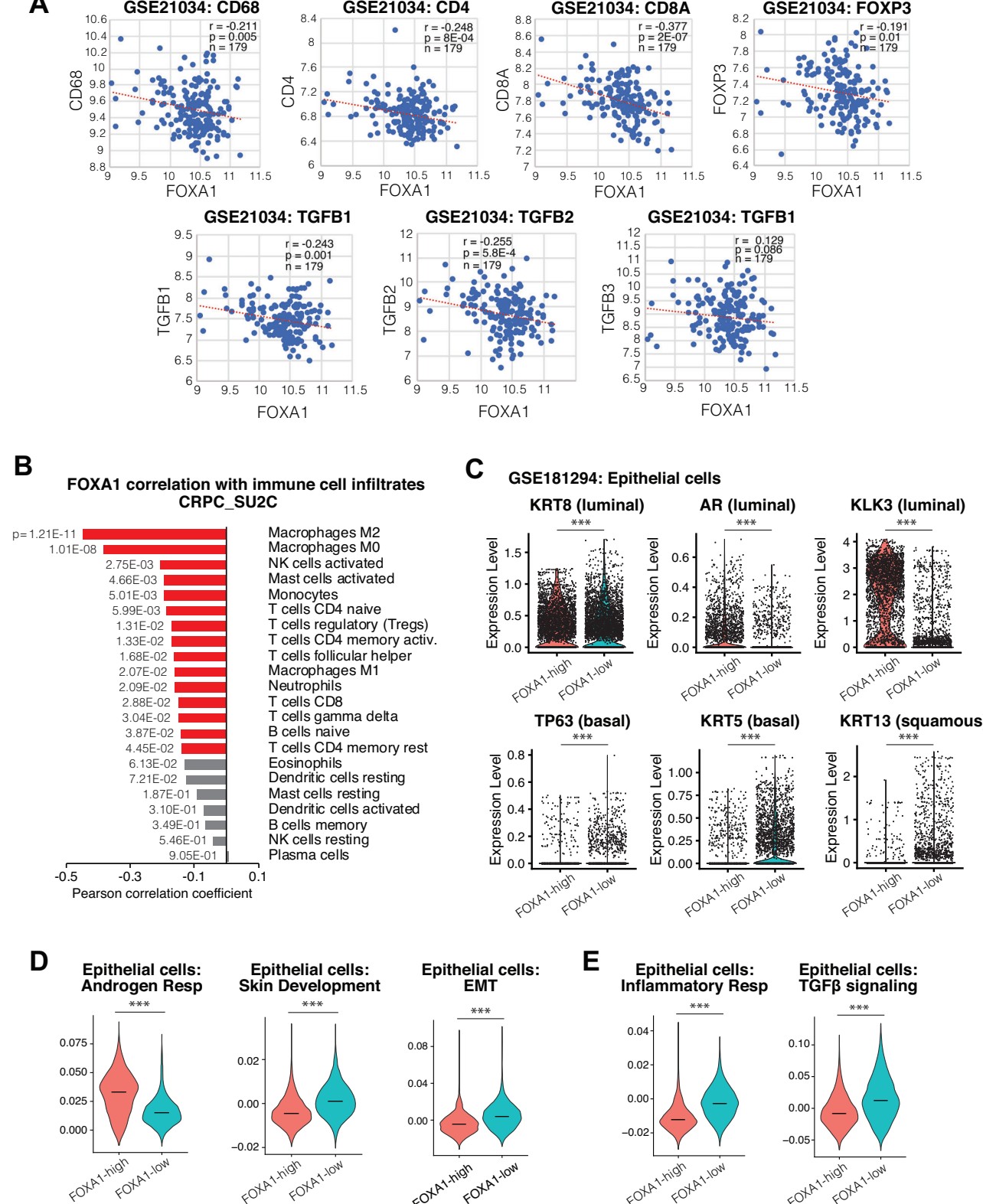

TG; Internal positive control R, GTC AGT CGA GTG CAC AGT TT; Tg=393 bp; internal control= 200 bp) (Supplementary Data 4).

## FFPE mouse tissue harvesting & processing

Mouse prostate tissues were harvested at desired timepoints and immediately placed in 10% formalin for fixation ~20–24 h, then stored in 70% ethanol until proceeding with paraffin processing and embedding. Tissues were paraffin-processed and embedded

through the Mouse Histology & Phenotyping Laboratory (MHPL) at Northwestern University. Tissue blocks were sectioned in house (5 um thickness).

## Hematoxylin and eosin (H&E) Staining

FFPE tissue sections were deparaffinized by 5 min in xylene (3x), followed by 5 min in 100% Ethanol (2x), 5 min in 95% ethanol, 5 min in 70% ethanol, and transfer to DI water. Slides were stained in Hematoxylin

**Fig. 7 | FOXA1 downregulation is associated with immunosuppression and epithelial lineage switch in human prostate tumors. A** FOXA1 correlation with major immune cell marker genes (*CD68, CD4, CD8A*, and *FOXP3*) and TGF-β cytokine genes (*TGFB1, TGFB2*, and *TGFB3*) in GSE21034 PCa dataset. (*n* = number of patient samples, r = Pearson correlation coefficient, and *p* = p-value.). **B** FOXA1 correlation with CIBERSORTx deconvolution estimated immune cell infiltration levels in CRPC SU2C patient dataset[61]. A two-sided Pearson correlation test using a t-distribution with n − 2 degrees of freedom. (*n* = 208 patient samples). **C** Violin plots depicting expression level of epithelial lineage related genes of interest *KRT8* (*p* = 9.06e-10), *AR* (p = 3.72e-153), *KLK3* (*p* < 2.2e-16), *TP63* (*p* = 4.74e-33), *KRT5* (*p* = 5.50e-135), *and KRT13* (*p* = 6.97e-110) within the epithelial population of PCa patients (GSE181294) stratified by FOXA1-low (n = 3687 epithelial cells) vs high (*n* = 3005 epithelial cells) expression. Statistical significance determined using ggbetweenstats, two-sided Mann-Whitney nonparametric test; ***p-value < 0.001. **D** Expression of Hallmark Androgen Response (*p* = 1.44e-283), Epithelial mesenchymal transition (EMT; *p* = 1.57e-152), and GO Biological Processes Skin Development (*p* = 1.82e-187) signature gene sets within epithelial populations in FOXA1-low (*n* = 3687 epithelial cells) vs high (*n* = 3005 epithelial cells) patients. (Horizontal black bar indicates median. Statistical significance determined using ggbetweenstats, two-sided Mann-Whitney nonparametric test; ***p-value < 0.001. **E** Expression of Hallmark TGF-β signaling (*p* = 8.30e-141) and Inflammatory Response (*p* = 0.00) gene sets within epithelial populations in FOXA1-low (*n* = 3687 epithelial cells) vs high (*n* = 3005 epithelial cells) patients. Statistical significance determined using ggbetweenstats, two-sided Mann-Whitney nonparametric test; ***p-value < 0.001.

(Hematoxylin Gill III MilliporeSigma 6506775) for ~40 s and immediately transferred to a beaker with running water for several minutes. Sections were then counterstained with Eosin (VWR Eosin Stain Premium 95057-848) for ~40 s. Next, Tissues were dehydrated by dips through 70% Ethanol, 95% Ethanol, and 100% Ethanol (2x), followed by 2 min in Xylene (3x). Finally, coverslips were mounted using permount (Fisher Chemical™ Permount™ Mounting Medium SP15-500).

### IHC Staining
Tissue sections were deparaffinized by dips in xylene for 5 min (3x), 100% Ethanol for 5 min (2x), 95% ethanol for 5 min, and 70% ethanol for 5 min. Tissue sections were transferred to TBS. Next, tissue sections were transferred into a slide container with pre-heated sodium citrate buffer in a pressure cooker for heat-induced antigen retrieval. Slides in the sodium citrate buffer were heated in the pressure cooker on a 15 min steam program. When complete, the slide container was placed into ice cold water for 1 h to cool. Next, Slides were rinsed 3 times x 5 min in TBS. A pap pen was then used to draw a circle around each tissue, slides were placed into a humidifier slide case, and 0.5% Triton X-100 was pipetted onto each tissue for tissue permeabilization for 15 min. Following tissue permeabilization, tissues were washed 3 times x 5 min in TBS. Next, we used Vector Lab Impress anti-Rabbit or M.O.M. (mouse-on-mouse) kit depending on primary antibody host and followed kit specific instructions. Finally, we counterstained with hematoxylin ~35–40 s and washed in a beaker with running water. Finally, we dehydrated the tissue by 3 min in 50% ethanol, 3 min in 70% ethanol, 3 min in 100% ethanol (2x), and 3 min in xylene (3x). Coverslips were mounted using permount. The following antibodies were used for IHC staining: Recombinant Anti-FOXA1 antibody [EPR10881] ab170933 Rabbit (1:1000), PTEN Cell Signaling Technology (D4.3) XP® Rabbit mAb #9188 (1:75), Recombinant Anti-Androgen Receptor antibody [ER179(2)] (ab108341) (1:1000), Anti-p63 Antibody (D-9) Santa Cruz sc-25268 Ms (1:250), Purified anti-Cytokeratin 8 Antibody Biolegend 904804 Ms (1:400), pSMAD2 (MilliporeSigma; AB3849; 1:1000), F4/80 (CST 70076; 1:500), CD4 (CST 25229; 1:100), and CD8 (CST-98941; 1:400).

### Single cell RNA-seq of whole mouse prostate tumors with 10X Genomics
Chromium Next GEM Single Cell 3′ Kit v3.1 (Dual Index) was used for single-cell RNA-seq analysis of whole prostate tumor tissues (P12, PF12, P18, PF18, XP, and XPF). Working space and tools were prepared with 70% ethanol and RNase-Zap (Invitrogen). Mouse prostate tissue was resected and immediately processed for downstream scRNA-seq analysis. Prostate tumors were cleaned to remove excess fat, seminal vesicles, and vans deferens. Tissue was dissociated using the Worthington-Biochem papain dissociation kit (according to manufacturer instructions). In brief, clean mouse prostate tissue was minced into small pieces and added to papain solution containing 20 units/ml papain, 0.005% DNase, and 10 uM Y-27632 in EBSS (Earle's Balanced Salt Solution). The tube was incubated in a water bath at 37 °C, shaking every 5–10 min, for 1 h. The tissue was then triturated with a wide bore tip 1 ml pipet and filtered through a 70um cell strainer. The cell suspension was centrifuged at 150–300 g for 5 min, the supernatant was aspirated, and the cell pellet was resuspended in a DNase dilute albumin-inhibitor solution consisting of 850 ul EBSS, 100 ul reconstituted albumin-ovomucoid inhibitor solution, and 0.005% DNase. A discontinuous density gradient was prepared to separate membrane fragments from dissociated cell pellet by layering the cell suspension on top of 5 ml albumin-inhibitor solution and centrifuging at 70 g for 6 min. Supernatant was discarded, and the pellet was resuspended in TrypLE + 10 uM Y-27632 and incubated at 37 °C for 15 min, pipetting the cell suspension every 5 min. The suspension was passed through a 40 um strainer (Flowmi) and TrypLE was inactivated by the addition of RPMI + 10% FBS. Following dissociation, cell count, and viability were measured by Trypan blue using Cell Countess II to ensure single cell suspension and viability > 80%. If doublets/multiplets remained, the TrypLE dissociation steps were repeated. Finally, cells were centrifuged at 300 g for 5 min and resuspended in RPMI + 10%FBS for loading into the 10X Chip G, with a targeted cell recovery of 4000–5000 cells. A special thanks to Dr. Alexander Misharin for generously sharing their lab's Chromium Controller for GEM generation. The Biorad C1000 Touch or MyCyler Thermal Cycler system was utilized for protocol steps requiring thermal cycler incubation, depending on volume requirements. Amplified cDNA and final libraries were analyzed by Qubit dsDNA HS Assay and Agilent BioAnalyzer (NUSeq core). Libraries were sequenced on Illumina HiSeq2000 PE150 through FSU or NovaSEQ-SP-100 through UC.

### Single cell RNA-seq of CD45+ sorted mouse prostate tumors with PIP-seq
PIPseq T20 3′ Single Cell RNA Kit v3.0 was used for single-cell analysis of CD45+ tumor infiltrating immune cells (samples P15 and PF15). Mouse prostate tumor tissues were dissociated as described in the 10X scRNA-seq methods. For live/dead staining, cells were stained with Zombie Aqua Live/Dead stain (Biolegend Zombie Aqua™ Fixable Viability Kit # 423101, 1:500) for 30 min at 4 °C. For CD45 staining, Fc block (TruStain FcX™ PLUS Antibody # 156603, 1:150) was applied for 5–10 min, followed by CD45 + APC stain (Biolegend APC anti-mouse CD45 Antibody # 147707, 1:100) for 30 min. Cells were washed and resuspended in 2%FBS + 25 mM HEPES PBS for cell sorting. Live CD45+ cells were sorted into 20%FBS collection media using the FACS Aria II through the Emory Pediatrics' + Winship Flow Cytometry Core. After sorting, cells were washed with PIP-seq cell resuspension buffer and continued with the PIP-seq protocol as described in the user guide. Amplified cDNA and final libraries were analyzed by Qubit dsDNA HS Assay and Agilent BioAnalyzer (Emory Integrated Genomics Core). Libraries were sequenced on Illumina HiSeq2000 PE150 through FSU.

### Single-cell RNA-Seq Analysis
For 10x Genomics single-cell RNA-seq, libraries were aligned to hg38 using CellRanger[75]. Low-quality cells with low UMI counts and high mitochondrial content within each mouse were filtered out. The individual samples were normalized using SCTransform v2, and then

integrated with castrated mice samples using the Seurat V3 integration pipeline[76]. Clustering was performed at resolution 0.2, yielding 17 clusters (0–16). Clusters were subsequently annotated based on cell type marker gene expression, with several clusters combined based on similar marker gene expression and proximity in the UMAP (Figure S2A–D). Visualizations and differential gene expression analysis utilized LogNormalized data. For PIP-seq single-cell RNA-seq, libraries were aligned using PIPseeker to hg38 and sensitivity 5 results were chosen. Further removal of low-quality cells and integration were performed in the same manner as 10x single-cell RNA-Seq. For the trajectory analysis, only epithelial cells were selected from the integrated object and re-integrated to create a new UMAP. Trajectory graph and pseudotime calculation was performed with this UMAP using Monocle3. Dittoseq was utilized to visualize cell type proportions[77]. Differentially expressed (DE) genes between P and PF conditions were determined utilizing Seurat FindMarkers() function and the MAST package for DE testing[78], with |log2 fold change| cutoff > 1. The overlap of PF upregulated or downregulated genes with GO BP (biological processes) or Hallmark gene sets in MSigDB were determined[42,79,80]. Seurat AddModuleScore() function was utilized to evaluate module scores for gene expression programs. Signature gene programs utilized to evaluate immune cell phenotypes are found in Supplementary Data 1. Signature gene programs were modified from existing signature gene sets in the literature, for analysis of mouse or human data[21].

## CellChat analysis of 10x scRNA-seq data

Cell-cell interactions were evaluated using the CellChat v2 R package[48,49], which estimates cell-cell interactions based on the expression level of a ligand and its corresponding receptor on the cells. The CellChat ligand-receptor database is informed by literature knowledge and includes subsets for Secreted Signaling L-R pairs (cytokines, chemokines, growth factors, etc.), ECM-Receptor, Cell-Cell Contact, and Non-protein signaling. Data analysis was performed following the tutorial for comparison analysis of multiple datasets (https://github.com/jinworks/CellChat). Only the Secreted Signaling subset was utilized to focus our analysis on evaluating the role of FOXA1 in regulating cytokine signaling interactions. Population size was set to true to take the cell proportions into account in the probability calculation.

## Spatial Transcriptomics analysis with 10x Genomics Visium

Spatial transcriptomic analysis was performed using the 10x Genomics Visium platform on tumor sections from P12, PF12, P18, and PF18 mice (one biological pair per time point; total $n = 2$ pairs). The limited number of replicates reflects the cost and technical complexity of the assay. Mouse prostate tissues were harvested at desired timepoints and formalin fixed for ~20–24 h. Tissues were paraffin-processed and embedded by the Mouse Histology & Phenotyping Laboratory (MHPL) at Northwestern University. Tissue histology was analyzed prior to Visium experiment by H&E and IHC staining to confirm representative samples were chosen and to select regions for tissue scoring. Tissue blocks were scored, sectioned (5 μm) and placed onto the Visium capture areas in lab, using RNaseZap treated tools/equipment. Visium slides were then immediately sent to the NUseq core for sample processing according to the 10x Visium for FFPE guide. Libraries were sequenced through NUseq core using Novaseq6000 Sequencer.

## TESLA analysis of Visium data

We utilized a machine learning method, TESLA[47], to annotate cell type distribution om Visium spatial transcriptomics data. Briefly, TESLA takes the selected cell type marker genes as inputs, enhances the expression of cell-type-specific marker genes to the same pixel resolution as H&E images, and integrates these data to determine the abundance of target cell types. The method assigns each pixel a score between 0 and 1 that reflects the relative enrichment of a given cell type at that pixel across the whole slide. To identify regions enriched for a specific cell type, pixels with enrichment scores > 0.3 were classified as enriched for that cell type and used to quantify the corresponding area. Notably, because these scores represent relative enrichment rather than exclusive assignment, a pixel may have non-zero scores for multiple cell types, capturing their colocalization pattern. Cell type marker genes used for our analyses are summarized in Supplementary Data 2. The identified immune populations were overlaid with epithelial tumor regions to visualize immune infiltration within tumor areas.

## ChIP-seq of mouse prostate tumors

For ChIP-seq analysis of mouse prostate tissues, 40 mg of flash frozen 18wk timepoint P and PF prostate tumors ($n = 3$ per genotype) were utilized for the FOXA1 ChIP experiments and 20 mg for H3K27ac. Tissues were ground using agate mortar and pestle and homogenized using the BeadBug benchtop homogenizer. Tissues were then double cross linked with 2 mM DSG for 10 min, followed by 1% formaldehyde for 10 min, at room temperature. Crosslinking was quenched with 0.125 M glycine for 5 min at room temperature. Chromatin was fragmented using an E220 focused ultrasonicator (Covaris), 50 ng of Drosophila chromatin was added to each sample as spike-in DNA, and samples were pre-cleared with protein A agarose beads (Millipore) for 1 h. Then samples were incubated with the following antibodies overnight at 4 °C with rotation: FOXA1 ab23738 (5 μg; Abcam) and H3K27ac CST 8173S (1:100). 1 μg drosophila H3 antibody (Active motif Cat# 61686) was also added per sample. Then, protein A agarose beads were added and incubated for 2 h at 4 °C, followed by washing the beads with 1×dialysis buffer (2 mM EDTA, 50 mM Tris-Cl, pH 8.0) twice, and IP wash buffer (100 mM Tris-Cl, pH 9.0, 500 mM LiCl, 1% NP40, 1% deoxycholate) four times. Finally, protein-DNA complexes were eluted (50 mM NaHCO$_3$, 1% SDS), crosslinks were reversed, and DNA was purified using DNA Clean & Concentrator-5 kit (ZYMO Research). Total 10 ng ChIP DNA was used to prep library (NEB E7645L).

## ChIP-seq data analysis

Duplicate reads were identified and removed by Picard (v3.0.0) and the adaptor reads removal process was performed with Trimmomatic V0.39. The reads were aligned against the Mus musculus reference genome mm10 and the *D. melanogaster* reference genome Dmel A4.10 using Bowtie2 (v2.5.1). The spike-in normalization was performed by counting *D. melanogaster* reads in each sample to calculate a spike-in normalization factor, and down-sampling was performed by samtools (v1.17) by random down-sampling to the smallest spike-in counts. HOMER (v4.11) was utilized for ChIP-seq peak calling with default cutoff. Overlap of FOXA1 ChIP-seq peaks was determined using HOMER default setting. To evaluate enrichment of peak, the number of unique aligned reads overlapping each peak in each sample was calculated by subread (v 2.0.6). The differential peak enrichment was analyzed by R Bioconductor package DEseq2 at the indicated FDR-adjusted $p$-value and log$_2$fold-change cutoffs ($p_{adj} < 0.05$, |log$_2$fold-change| >1|). ChIP-seq heatmaps were generated with deepTools (v3.5.4) and show normalized read counts 5 kb before and after peak center. ChIP-seq peaks were assessed for annotations among genes near ChIP-seq peaks, assigning each peak to the TSS of nearest gene using HOMER. The R package clusterProfiler was used to perform enrichment analysis on Gene Ontology and Hallmark gene set from MSigDB[42].

## Flow cytometry and CyTEK analysis

The single-cell suspensions underwent stimulation with phorbol 12-myristate 13-acetate (50 ng/ml), ionomycin (5 μg/ml), and brefeldin A (10 μg/ml) to induce IFNγ and TNFα production. To block non-specific antibody binding to Fc receptors on immune cells, the stimulated

single-cell suspensions were incubated with anti-mouse CD16/32 antibody (2.4G2). Following this, live/dead dye and fluorophore-conjugated anti-mouse antibodies targeting cell surface markers were added. After PBS washing, cells were fixed with 1x Fixation/Permeabilization reagent (eBioscienceTM, 00-5223-56) for 20 min at room temperature. Subsequently, cells were washed twice with 1x Permeabilization/wash buffer (eBioscienceTM, 00-8333-56) and subjected to intracellular staining (ICS) with an antibody cocktail for 45 min at room temperature. Finally, cells were washed and suspended in phosphate-buffered saline (PBS) for acquisition using a three-laser CyTEK AURORA spectral flow cytometry or LSRII flow cytometry. Single staining of each antibody with corresponding fluorescence on splenocytes served as reference for spectral unmixing (CyTEK). Data analysis was performed using either the OMIQ web application (https://www.omiq.ai/) or the FlowJo application, according to previously described methods (PMID: 35671108; PMID: 38530357). Antibody information is listed in Supplementary Data 3.

### CIBERSORTx analysis of publicly available patient datasets
CIBERSORTx analysis[60] was performed on the SU2C mCRPC RNA-seq dataset[61] to impute cell fractions. The following parameters were used for the run: LM22 signature matrix[81], B-mode batch correction, absolute run mode, 100 permutations. Pearson correlation between *FOXA1* expression and immune infiltrate levels were determined.

### Peripheral blood mononuclear cells (PBMC) isolation
Blood samples were collected from patient donors with prostate cancer after receiving written informed consent under a protocol approved by the Institutional Review Board at the William S Middleton Memorial Veteran's Hospital, Madison, WI in accordance with the Declaration of Helsinki. Blood specimens were collected in vacutainer tubes (BD Biosciences, Franklin Lake, NJ, USA) with EDTA anticoagulant. Whole blood was diluted 1:1 with Hank's balanced salt solution (HBSS, Lonza Group, Basel, Switzerland) before being underlaid with 10 mL of Ficoll-Paque PLUS (GE Healthcare, Cat# 45-001-750) for gradient centrifugation. CD14+ monocytes were enriched from PBMCs using LS MACS columns following incubation with anti-CD14 magnetic beads (Miltenyi Biotec Inc., Bergisch Gladbach, North Rhine-Westphalia, Germany).

### Cell Culture & TGFB Receptor Inhibitor Treatment
Transfected PC-3 cells, control and *FOXA1* knockdown (pGIPZ-shCtrl and pGIPZ-shFOXA1 (target sequence: TGGACAGACTTCCAAGATG), obtained from Open Biosystems), were cultured in RPMI1640 media with L-Glutamine (CorningTM Thermo Fisher Scientific, Waltham, MA, USA), 10% FBS (GibcoTM, Thermo Fisher Scientific, Waltham, MA, USA), and 2% penicillin/streptomycin (HycloneTM, VWR, Radnor, PA, USA). 1 ug/mL Puromycin (GibcoTM, Thermo Fisher Scientific, Waltham, MA, USA) was added to the media to select for transfected cells and was refreshed with new media daily. PC-3 cell line was obtained from ATCC and periodically tested for potential mycoplasma contamination. Patient-derived monocytes were cultured in RPMI1640 media with L-Glutamine, 10% FBS, 5% Glutamax (GibcoTM, Thermo Fisher Scientific, Waltham, MA, USA), and 2% penicillin/streptomycin. Single cell suspensions of PC-3 cells were harvested at log phase and were seeded onto Stacks hydrogel wells at a concentration of 150k cells per mL. Using established protocols (PMID 36083096, 39414902, 40605246), CD14+ monocytes were differentiated into macrophages using 50 ng/mL colony stimulating factor 1 (CSF1) (RnD systems, Minneapolis, MN, USA), administered on Days 1 and 4, and were seeded in Stacks at a concentration of $3 \times 10^6$ cells per mL. PC-3 and patient-derived primary macrophages were cultured in Stacks by seeding as a monolayer on a collagen-fibronectin matrix, consisting of 79% collagen I (Advanced BioMatrix, Carlsbad, CA, USA #5005), 1.5% fibronectin (Sigma-Aldrich, Millipore Sigma, Burlington, MA, USA

#F1141) prepared according to manufacturer guidelines. Moisture was retained by storing Stacks inside of a humidifying chamber.

Prior to treatment and stacking, the PC-3 wells were washed twice with a 30-minute incubation using plain media to remove the puromycin. TGF-β receptor I inhibitor Galunisertib (LY2157299) at a stock concentration of 10 mM was diluted at a 1:2000 dilution in media to 5 μM solution. Treatment was applied for 24 h as a 20 μL of media with 5 μM Galunisertib or vehicle control on day 8.

### Stacks devices
Prior to culture, Stacks plates (Protolabs, Maple Plain, MN, US #1121-5161-007) were prepared by sonication in 100% isopropanol for 60 minutes, and air dried inside a BSC on a pre-sterilized rack for 10–20 min before rotating and drying for an additional 10–20 min. Stacks devices, 3D holders, NuncTMOmnitraysTM (Thermo Fisher Scientific, Waltham, MA, USA), and non-tissue culture-treated BioAssay dish (245 mm square; Corning Inc., Corning, NY, USA) were sterilized with 70% ethanol followed by 20 min UV light treatment on each side. Cells were plated in the Stacks device and then allowed to adhere or migrate through the matrix as applicable. When stacking devices, media was removed from the top and bottom leaving only a small volume of residual media to prevent gas bubble formation during stacking. Stack devices were placed in a three-layer humidifying chamber including a sterile sponge soaked in sterile ddH2O in a NuncTMOmnitrayTM(Thermo Fisher Scientific, Waltham, MA, USA).

### Nucleic Acid Extraction and Quantitative RT-PCR
Cells were removed from hydrogel matrix via application of 5 μL of 1 mg/mL collagenase I (Sigma-Aldrich, Millipore Sigma, Burlington, MA, USA) for matrix digestion prior to transferring single cell suspensions to 96-well plates for subsequent lysis containing 40 μl of lysis/binding buffer solution + 10 μL of LB washed beads per well. mRNA isolation was performed using InvitrogenTM DynabeadsTM mRNA DIRECT Kit (Thermo Fisher Scientific, Waltham, MA, USA). The lysate was washed with 100 μL Buffer A x 2 and 100 μL Buffer B x 1, then suspended in 16 μL of nuclease-free H2O (PromegaTM, Madison, WI, USA).

The mRNA elution sample was reverse transcribed using 4 μL SuperscriptTM IV VILOTM Master Mix (InvitrogenTM, Thermo Fisher Scientific, Waltham, MA, USA), according to manufacturer's directions using Bio-Rad C1000 TouchTM thermocycler (Bio-Rad Laboratories, Hercules, CA, USA). The subsequent cDNA solution (6.3 μL) was then amplified using 12.5 μL TaqManTM PreAmp master mix (Applied BiosystemsTM, Thermo Fisher Scientific, Waltham, MA, USA) plus 6.25 μL of a Primer mix. The primer mix contained 20 μL of each available primer and TE buffer to bring the mix up to 2000 μL final volume. 10 cycles of pre-amplification were performed according to manufacturer's directions, and the final reaction volume was then diluted by adding 200 μL nuclease-free H2O. For TaqManTM qPCR assays, 2.5 μL of diluted cDNA template was mixed with 5 μL TaqMan FastTM master mix (Bio-Rad, USA), 0.5 μL TaqManTM Gene Expression Assay C-C motif chemokine ligand 2 (*CCL2*) (HS00234140_m1), transforming growth factor beta 3 (*TGFB3*) (Hs01086000_m1), forkhead box A1 (*FOXA1*) (Hs04187555_m1), macrophage migration inhibitory factor (MIF) (Hs00236988_g1), and housekeeping genes: ribosomal protein lateral stalk subunit P0 (*RPLP0*) (Hs00420895_gH) and RNA polymerase II subunit A (*POLR2A*) (Hs00172187_m1) (Life Technologies, USA)) and 2 μL nuclease-free water. Each reaction was amplified in duplicate using a QuantStudioTM 5 Real-Time PCR system via the Comparative CT Fast protocol available on the machine. Analysis was performed using the Thermo Fisher ConnectTM online application.

### Confocal microscopy analysis
For confocal microscopy analysis, cells were then fixed at their location within the collagen-fibronectin matrix (eBioscience, Thermo Fisher

Scientific, Waltham, MA, USA; # 00-5521-00), washed, and then stored at 4 °C. Staining was performed via overnight incubation at 4 °C of a 1x PBS/10% FBS solution comprised of 20 mM Hoechst 33342 (1:250) and anti-CD14 (BD Biosciences, USA # 561708) (1:50). Spinning disk confocal microscopy was performed at the University of Wisconsin–Madison Biochemistry Optical Core (RRID:SCR_023952), which was established with support from the University of Wisconsin–Madison Department of Biochemistry Endowment, at 10x magnification with 10 um z-layers, for a total of 101 layers. Images were analyzed with NIS-Elements software (RRID:SCR_014329; Nikon Instruments, Melville, NY, USA). Individual cells were identified using the 3D spot detection function via a recipe generated with the General Analysis 3 (GA3) module in the Hoechst channel to identify each nucleus. Migrated cells are classified as cells found below 200 μm, which is the approximate depth of the biomatrix meniscus within the Stacks well.

### PF tumor allografts

To establish PF tumor allografts, prostate tumors from 15 to 18 week old PF transgenic mice were harvested, minced, and subcutaneously engrafted in the right flank of immunodeficient NSG mice to expand. Tumors that successfully engrafted in NSG mice (reached at least 300 mm³) were harvested, minced and subcutaneously engrafted into the right flanks of three pairs of NOD-SCID mice, which lack T and B cells but retain macrophages, for treatment with vehicle or 75 mg/kg TGF-β receptor I inhibitor Galunisertib (LY2157299) twice daily for three weeks.

### Statistics & Reproducibility

The statistical significance of differences in gene expression between conditions in the scRNA-seq data was performed using the ggbetweenstats function within the ggstatsplot package, with a two-sided Mann-Whitney nonparametric test[82]. Statistical analyses for Cytek spectral flow data, tumor size differences, mouse survival, qRT-PCR, and co-culture assays were performed using GraphPad Prism. Descriptions of the statistical tests used in the experiments can be found in the corresponding figure legends. All H&E and IHC images are representative of independent biological replicates as indicated in the figure legends (≥ 3 per genotype). No statistical method was used to predetermine sample size. No data was excluded from the analyses. The investigators were not blinded to allocation during experiments and outcome assessment. All animal experiments and human analyses included only male subjects, as the prostate is a male-specific organ. Consequently, sex was not included as a variable in the study design or statistical analysis.

### Reporting summary

Further information on research design is available in the Nature Portfolio Reporting Summary linked to this article.

## Data availability

All scRNA-seq, ChIP-seq, and Visium spatial transcriptomics data generated for this study have been deposited in the Gene Expression Omnibus (GSE282726) at https://www.ncbi.nlm.nih.gov/geo/query/acc.cgi?acc=GSE282726. Previously published microarray data for LNCaP cells with FOXA1 KD and rescue with WT or mutant FOXA1 constructs is deposited in GSE128882[13] (https://www.ncbi.nlm.nih.gov/geo/query/acc.cgi?acc=GSE128882). Previously published data for FOXA1 ChIP-seq in LNCaP cells is deposited in GSE55007[6] (https://www.ncbi.nlm.nih.gov/geo/query/acc.cgi?acc=GSE55007). Publicly available human PCa datasets analyzed in this study can be found at GSE181294[21] (https://www.ncbi.nlm.nih.gov/geo/query/acc.cgi?acc=GSE181294) and GSE21034[59] (https://www.ncbi.nlm.nih.gov/geo/query/acc.cgi?acc=GSE21034). Source data are provided with this paper.

## Code availability

The code for next-generation sequencing analysis performed in this paper has been uploaded to https://github.com/JYULAB/FOXA1_immune_project.

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

## Acknowledgements

NIH/NCI training grant T32CA009560 (to L.B.), F31CA271826 (to L.B.), R01CA257446 (to J.Y.), R01CA275193 (to J.Y., J.C.Z.), P50CA180995 (to J.Y.), and Prostate Cancer Foundation Challenge Award 2017CHAL2008 (to J.Y.). We thank all Yu lab members for their helpful discussions and suggestions. We thank the Northwestern University Immunotherapy Assessment core for their assistance with Cytek spectral flow experiments. We thank Dr. Jeffrey Sosman for helpful discussions and Dr. Jennifer Wai from the NUSeq core at Northwestern for helping perform 10x Genomics Visium experiments. We also thank Dr. Alexander Misharin for generously sharing their lab's Chromium Controller for our 10x Genomics scRNA-seq experiments. We also thank the Emory Pediatrics + Winship Flow Cytometry Core for their assistance with cell sorting, as well as the Emory Integrated Genomics Core (EIGC). We also thank the Northwestern University Mouse Histology and Phenotyping Laboratory Core for their tissue processing and embedding services and the Center for Advanced Microscopy/Nikon Imaging Center (CAM) for the use of their widefield microscope. We thank the University of Wisconsin–Madison Biochemistry Optical Core for use of shared equipment and excellent technical support. We would also like to thank Emma Recchia and Vilena Maklakova for their technical support. This project was supported in part by Merit Review Award I01 CX002479 to D.K. from the United States Department of Veterans Affairs Office of Research and Development, Clinical Sciences Research and Development Service, and the P30CA014520-UW Carbone Cancer Center Support Grant (CCSG). The information presented here solely represents the views of the authors and does not represent the views of the United States Government or the Department of Veteran Affairs.

## Author contributions

J.Y. and L.B. conceived the project and designed the experiments. H.S. and W.X. assisted with mouse breeding, tissue collection, and tissue sectioning. W.X., H.S., and L.B. performed IHC and H&E staining. Y.X. and S.S. evaluated tumor metastasis in the mice. L.B. performed scRNA-seq experiments. L.P. conducted ChIP-seq experiments. J. Huang and J. Hu conducted spatial transcriptomics bioinformatics analysis. V.K., Q.C., J.C.Z., L.B., and J.Y. conducted other bioinformatic and statistical analyses. J.F. and P.X. performed and analyzed the Cytek spectral flow experiments, overseen by B.Z. D.J.D. provided the Foxa1 KO mouse and helpful discussions. M.B. and D.K. conducted in vitro co-culture assays. X.Y. consulted on mouse tumor histology. X.L., B.Z., and S.A.A. consulted on the project. L.B. and J.Y. wrote the original manuscript.

## Competing interests

The authors declare no competing interests.
