## [Transparent Peer Review file · Nature Communications]

FOXA1 loss drives basal/squamous de-differentiation of prostate cancer and induces an immunosuppressive tumor microenvironment

Corresponding Author: Dr Jindan Yu

Version 1:

Reviewer comments:

Reviewer #1

(Remarks to the Author)

Brea and colleagues describe a new genetically engineered mouse model of PCa bearing the concomitant prostate specific loss of Pten and Foxa1. By coupling classical histopathological studies with approaches of single cell sequencing and spatial transcriptomics, the authors describe a critical oncosuppressive role of Foxa1 in Pten-loss driven mouse prostate tumorigenesis. Their findings highlight two major phenotypes associated with the loss of Foxa1: the skewing of the luminal lineage versus the basal/squamous lineage of the prostatic epithelium combined with the generation of an immunosuppressive tumor microenvironment.

The manuscript is well-written, the topic is clinically relevant, and the results are interesting.

Below few major aspects that Authors should carefully consider and address:

1. To better understand the relevance of the genetic model established in mirroring human disease, the Authors should define the incidence of PTEN-FOXA1 double null condition in human prostate cancer considering primary, metastatic and castration resistant stages of the disease.
2. Absence of basal cell markers such as TP63 and CK5 is generally coupled to H&E staining in PCa diagnosis. Authors should indicate if and how the expansion of TP63 staining (and maybe also CK5) recapitulates specific genetic or histological subtypes of human PCa.
3. Figure 1 nicely describes the histology of the different lobes of the mouse prostate in the wild type, Pten-null and Pten-Foxa1 double null model, while 1F defines the lethality of the PF condition compared to P only. Although bladder obstruction may most likely be the reason for lethality, the authors should present a careful analysis of proximal lymph nodes, distant organs (e.g., liver and lung), and bone (e.g., femurs) to exclude or perhaps include metastatic disease progression, which, given the paucity of metastatic GEMMs for PCa, would greatly increase the relevance of the study.
4. Bulk and scRNAseq studies demonstrate a different immune profile of P and PF prostate tumors with PF described as more immune suppressive/exhausted than P. A critical aspect of these analyses is related to the substantial different stage of P and PF tumors. As clearly described in Figure 1, PF tumors are much more aggressive and advanced than P tumors at the same time points. Therefore, it is difficult, if not impossible, to conclude whether it is the loss of Foxa1 or the stage of the tumor that defines immune suppression/exhaustion. These analyses should be performed on histopathological similar -no age matched- P and PF tumors (i.e., P24 vs PF12).
5. Based on the data shown in Figure 4 immune cells appear to share common areas of PF tumor. Are these areas characterized by specific CAFs/extracellular matrix conditions. Fibroblasts are weirdly neglected in this work.
6. Figure 4 describe the spatial distribution of immune cells into the diseased prostate gland of P and PF mice. Are the different percentages described an average of n=? different P and PF tumors or the comparison of just a single P versus a single PF tumors. Are the data consistent across different set (i.e 100 um/10 sections away from those in Figure 4) of sections of the same couple of P and PF tumors?

7. Figure 5 describes the ability of PF tumors to generate an immune suppressor tumor microenvironment by modulating extracellular signalling. However, the conclusions are completely inferred from the differential RNA expression profiles, which is not sufficient to support the conclusion. Intratumor quantification of the most interesting factors through MS or ELISA methodologies should be included. As well, the immune suppressor role of specific factors (at least one) should be functionally demonstrated in vivo through the injection of specific inhibitors. Interestingly, based on panel B PF tumors express very low levels of Activin (Activin A?) compared to P tumors. Activin A has been described plays a very important role in controlling prostate tissue homeostasis in mouse adult prostate progenitors by promoting quiescence (Cambuli et al., 2022 EMBO Reports). Organoids that were able to spontaneously overcome the barrier by reducing Activin A expression/secretion experience genomic instability and generate CK8/CK5 double positive hyper proliferating dysplastic lesions when transplanted orthotopically into the prostate of adult wild type syngeneic mice. Authors should comment on the possible implication of reduced Activin expression in PF versus P prostate tumors.

8. A recent paper (De Felice et al., 2024) describes the role of Retinoic Acid signaling in promoting luminal lineage of mouse prostate progenitors by promoting Foxa1 expression. Lack of RA-Foxa1 activity as well as specific mutant forms of FOXA1 associated with human PCa determine AR deregulation and a basal/squamous commitment of prostate progenitor cells. Authors should carefully discuss this paper in their manuscript.

(Remarks on code availability)

Reviewer #2

(Remarks to the Author)

FOXA1 is a critical gene in prostate cancer development and therapeutic resistance, with functions that appear to be highly context-dependent. It is frequently mutated and overexpressed in prostate cancer, rarely deleted, but appears to be downregulated in castration-resistant prostate cancer (CRPC). Its primary role is as a pioneer factor for the androgen receptor (AR), and it has been implicated in AR-independent CRPC evolution through mechanisms such as cellular plasticity.

This complexity has led to seemingly conflicting findings, underscoring the multifaceted role that FOXA1 plays in prostate cancer pathogenesis. In this context, the generation and characterization of the PTEN-FOXA1 prostate knockout model presented in this manuscript is of considerable interest. The authors have previously investigated FOXA1 downregulation in human cells, primarily using shRNA, observing reduced cell proliferation in hormone-sensitive lines and upregulation of the epithelial-to-mesenchymal transition (EMT) program.

Interestingly, in this mouse model, co-deletion of PTEN and FOXA1 results in an aggressive prostate cancer phenotype with pronounced basal and squamous features and an immune infiltrate rich in myeloid cells. This suggests a potentially immunosuppressive tumor microenvironment characterized by dysfunctional T cells. Although basal/squamous prostate cancers are relatively rare in patients, I find this model highly intriguing, as it aligns with reports suggesting that this phenotype—more commonly observed in mice—may represent a plastic intermediate state in patients.

The authors employ a multimodal characterization of these tumors, which may have broader implications beyond the specific role of FOXA1. This is a significant strength of the manuscript, and I believe the study would benefit greatly from additional analyses and functional experiments.

Major Points:

Castration Studies

FOXA1 downregulation occurs later in prostate cancer progression, particularly during CRPC evolution, as noted by the authors. Characterizing these models post-castration—whether through surgical means or degarelix administration—is essential to fully demonstrate their relevance. The single-cell RNA sequencing (scRNA-seq) data indicate that the luminal compartment remains substantial. The impact of castration on these cells, as well as on the immune microenvironment, may influence the cancer phenotype and progression.

Metastatic Potential

Consistent with their previous studies in human cell lines, the authors report EMT upregulation upon FOXA1 deletion in these models. This raises an important question: do these tumors develop metastases? It is unclear whether the authors have investigated this aspect. Analyzing metastatic spread in both intact and castrated mice would provide a more comprehensive characterization of the model.

Addressing Discrepancies in Tumor Immune Microenvironment (TIME) Analysis

The authors provide an extensive characterization of the TIME using multiple single-cell resolution technologies, including scRNA-seq, PIP-seq, Cytex spectral flow cytometry, and spatial transcriptomics. However, while scRNA-seq and spatial transcriptomics reveal prominent macrophage and monocyte infiltration, flow cytometry identifies polymorphonuclear myeloid-derived suppressor cells (PMN-MDSCs) as the dominant immune cell type. PMN-MDSCs are of great interest in prostate cancer, as they play a crucial role in immune suppression in both mouse models and patients. The discrepancy between methods raises concerns. One possible explanation is methodological bias introduced by cell dissociation and

tissue preparation. On the other hand, the flow cytometry panel used may not be optimal for distinguishing MDSCs subsets. The standard antibody combination for M-MDSCs and PMN-MDSCs includes Ly6C (HK1.4) and Ly6G (1A8). However, the authors appear to have used Ly6C (HK1.4) in combination with the Ly6G/C antibody (RB6-8C5). When this antibody is used instead of a Ly6G-specific antibody such as 1A8, it stains both M-MDSCs and PMN-MDSCs, resulting in the majority of CD11b-positive cells being positive for this staining, as shown in Fig. S3. This could potentially lead to an overestimation of the number of PMN-MDSCs. Revisiting the staining strategy including additional validation with alternative antibodies is recommended.

Functional Experiments

The manuscript is largely descriptive. The authors acknowledge in the discussion that targeting cytokines such as TGF- β could influence immune checkpoint inhibitor (ICI) responses. This reasoning could also be extended to the observed TIME components. Performing a cell depletion assay or cytokine-targeting experiments—even without ICI combination—would provide functional evidence for the importance of at least one of these TIME components.

Additional Major Considerations:

Throughout the manuscript, including the title and discussion, the authors repeatedly emphasize how their data support the role of FOXA1 in prostate cancer progression through its regulation of prostate epithelial cell lineage identity and the tumor immune microenvironment. While the data may be consistent with this hypothesis, the manuscript lacks direct evidence—either genetic (e.g., lineage tracing) or mechanistic (e.g., ChIP-seq)—to prove it. Alternative explanations cannot be ruled out, making this the weakest aspect of the study.

To address this, I suggest two possible courses of action:

- The authors could revise the text to reduce emphasis on this aspect, ensuring that the conclusions are appropriately cautious.
- If the editors and authors believe this hypothesis is central to the study, additional experiments are necessary to provide stronger evidence.

Determining the Luminal or Basal Origin of the Tumor

The initial study describing the PB-Cre mice used in this study reported that "there was no evidence of Cre expression in the basal cells of any of the prostate lobes examined" (Wu et al., *Mechanisms of Development*, 2001). However, subsequent studies have shown that this model is not luminal-specific and that Cre is expressed in both basal and luminal epithelial cells (Wang et al., *PNAS*, 2006 [<https://doi.org/10.1073/pnas.051065210>]). Thus, to support the claim that FOXA1 drives dedifferentiation, the authors should investigate the tumor cell of origin in greater detail. The current data do not exclude a basal origin, and the two selected time points (12 and 18 weeks) do not clearly demonstrate a phenotypic switch. To determine whether luminal or basal cells initiate oncogenic transformation in this model, several approaches are available:

- Lineage tracing: This can be achieved using transgenic mouse models with CreERT2 systems more specific to either basal (e.g., CK5-CreERT2, TP63-CreERT2) or luminal (e.g., CK8-CreERT2, NKX3.1-CreERT2) epithelial cells, which would be more appropriate than the PB-Cre model.
- In vitro modeling: Isolating the different prostate epithelial cells (e.g., as organoids), transforming them genetically, and studying their oncogenic potential could provide more direct mechanistic insights. Performing similar experiments in human cells would further strengthen the study.

Chromatin Immunoprecipitation (ChIP)

The authors acknowledge that FOXA1's role is highly context-dependent. While the upregulation of EMT is consistent with previous findings in human cancer cell lines, the observed basal/squamous phenotype and cytokine signaling are novel insights specific to this model. To establish FOXA1's role in this process, ChIP-seq analysis in the relevant cell population is crucial. This would clarify:

- Whether FOXA1 directly regulates luminal differentiation.
- Whether the observed immunosuppressive TIME is a result of direct transcriptional repression of cytokines by FOXA1 or merely a consequence of the basal/squamous PCa phenotype, which may have its own distinct transcriptional network and immune landscape.

In conclusion, I see significant value in publishing a study describing the characterization of this PTEN/FOXA1 prostate cancer model, which represents a rare but intriguing subtype of prostate cancer observed in patients. However, I would recommend a more comprehensive characterization, including castration of the mice—either surgically or through degarelix treatment—as the genetic alterations in this model are intended to mimic castration-resistant prostate cancer. Additionally, an analysis of potential metastases should be included.

While I understand the inclination to link FOXA1 to cellular plasticity and propose it as a direct regulator of the tumor immune microenvironment through transcriptional control of the chemokine/cytokine milieu, this interpretation remains speculative. Although the characterization may suggest such a role, the study does not provide definitive proof. If the authors wish to pursue this hypothesis—which I agree is of considerable interest—they should conduct a combination of genetic, molecular, and functional analyses to substantiate their claims. Otherwise, they should adjust the manuscript's tone and conclusions accordingly.

(Remarks on code availability)

Bioinformatic analysis is not my area of expertise.

Reviewer #3

(Remarks to the Author)

Overview:

This manuscript (ms) explores the tumorigenic effects of *Foxa1* depletion in prostate cancer by comparing single-cell and spatial transcriptomic profiles between *Pten*-null transgenic mice (P) and *Pten/Foxa1* double-knockout mice (PF). Comprehensive bioinformatics analyses were conducted using various software tools to support the study's findings. Overall, this manuscript is well-written, with data and analysis results clearly presented. However, several issues need to be addressed.

Major Comments:

1. The study primarily analyzes four samples: P12, PF12, P18, and PF18 — representing two conditions at two different time points. A key limitation of the study is the lack of biological replicates for each combination of condition and time point. This is particularly concerning for the spatial analysis using TESLA, as different tissue structures of replicate samples could lead to varying conclusions.
2. In the scRNA-seq integration analysis, there seem to be six samples in total: P12, PF12, P18, PF18, and two castrated samples (XP and XFP). If the two castrated samples are indeed part of the integration analysis (as indicated in the analysis code), their absence from the UMAP visualization is unclear. Furthermore, there is no mention or representation of these castrated samples in any analysis or figures throughout the ms.
3. In the scRNA-seq integration analysis, several cell clusters co-express luminal and basal genes. For instance, the small basal (red) cluster in the center of Fig.2A shows high *Krt8* expression in Fig.2E, and similar patterns are seen in the "basal squamous" and "basal proliferating" clusters in Fig.S2A and S2B. Could these clusters represent doublets?
4. What does the UMAP plot in Fig.S2D look like when coloured by sample? Providing this visualization would help clarify sample-specific clustering and potential batch effects.
5. In the pseudotime analysis, both the "Basal proliferating" and "Tff3 Lum" clusters are assigned a zero pseudotime (Fig.2D). Does this imply that these clusters were excluded from the Monocle3 pseudotime trajectory analysis? The UMAP also suggests that luminal cells are closer to basal cells than basal proliferating cells, which seems counterintuitive.
6. How was the starting point chosen for the Monocle3 pseudotime analysis? Have the authors tested the robustness of their results by selecting different starting points — for example, the basal cluster on the left side of the UMAP?
7. The TESLA tool was used to annotate Visium data at the pixel level, but it appears that individual pixels do not have unique cell-type annotations. For example, in Fig.4A (PF18), there is substantial overlap between luminal and basal cells. Does this suggest that both luminal and basal cells occupy the same H&E image pixels? This ambiguity complicates the interpretation of cell-type proportions, as seen in Fig.4B, where macrophages, monocytes, and T cells together constitute over 80% of the tissue — a proportion unlikely to reflect the true composition.
8. Stromal cell populations are not clearly presented in the Visium profiles. Are stromal cells present in the analyzed tissues, and if so, where are they located?

Minor Comments:

1. Consistent colours for cell-type granularity across Fig.S2D and S2E would improve visual clarity and aid in the interpretation of results.

(Remarks on code availability)

Version 2:

Reviewer comments:

Reviewer #1

(Remarks to the Author)

Thanks to the authors for addressing the main issues.
The work is now suitable for publication.

(Remarks on code availability)

Reviewer #2

(Remarks to the Author)

The authors have carefully and thoroughly addressed all of the points raised in my previous review. The vast majority of the additional experiments requested have been performed, and the resulting data strengthen the overall conclusions of the study. Importantly, these new analyses have improved both the clarity and the robustness of the manuscript. I am pleased to see that the manuscript represents a substantial improvement over the original submission.

(Remarks on code availability)

Reviewer #3

(Remarks to the Author)

I appreciate the authors' efforts in addressing the comments. Most concerns have been adequately addressed. However, several important issues remain.

Major comment #3:

There is no clear description of doublet removal in the data analysis pipeline. The only mention of doublets appears in the TrypLE dissociation step during sample preparation, which does not guarantee the removal of all doublets for the downstream analysis. Proper doublet detection and filtering should be performed computationally during data processing, using established tools such as scDbtFinder or DoubletFinder.

Continuation of Major comment #3:

The current cell-type annotation is unconvincing. In Fig S2A, two clusters that are far apart from each other on UMAP are both annotated as basal cells (and similarly for the two well-separated Spink1 luminal clusters). According to the code on GitHub, clustering at a resolution of 0.2 yields 17 clusters (0–16). It is unclear why multiple, transcriptionally distant clusters (e.g., clusters 1, 5, and 15) are assigned the same basal cell identity. What marker genes distinguish these clusters (e.g., cluster 1 vs 5)?

Furthermore, the UMAP from the epithelial integration analysis (in the monocle3 analysis) suggests the presence of two transcriptionally distinct basal cell subsets (as well as two distinct Spink1 luminal subsets). These observations indicate meaningful biological heterogeneity that is not sufficiently explained by the current annotations. The rationale for assigning identical cell-type labels to these distinct clusters should be clearly justified, and the annotations revised if necessary.

Major comment #7:

Regarding the statement "... but rather % of captured spots in the tissue section with high probability of containing the indicated cell type," the definition of "high probability" is unclear. What probability threshold is used to classify a pixel as containing a given cell type? Or what exactly is the criterion for counting pixels for each cell type? While it is correct that Visium spots do not have single-cell resolution and may contain multiple cells, this argument does not apply at the pixel level, as image pixels are substantially smaller than individual cells and should therefore mostly correspond to a single cell type.

Additional point related to Major comment #7:

In the statement "... we believe the vast overlap of luminal and basal-like regions identified in the PF tumours could be indicative of a mixed luminal–basal phenotype and lineage plasticity, as observed in our scRNA-seq analyses as well," it is unclear which specific scRNA-seq clusters in the integration analysis represent this "mixed luminal–basal phenotype." The authors should explicitly identify the relevant clusters and provide supporting evidence for this interpretation.

Minor comment: It might be better to include Fig S2C in Fig 2D.

(Remarks on code availability)

The code on the GitHub is well organized and easy to follow.

Version 3:

Reviewer comments:

Reviewer #3

(Remarks to the Author)

I appreciate the authors' additional analyses addressing all the concerns, which have substantially strengthened the manuscript.

(Remarks on code availability)

Response to Referees

Manuscript Title: FOXA1 loss drives basal/squamous de-differentiation of prostate cancer and induces an immunosuppressive tumor microenvironment

Manuscript ID: NCOMMS-25-07398A-Z

Dear Reviewers,

Thank you for your careful evaluation of our manuscript and for the thoughtful and constructive comments provided. We have carefully considered all suggestions and revised the manuscript accordingly. Below, we present a point-by-point response to each comment, detailing the changes made and where they appear within the revised manuscript. All changes are highlighted in yellow in the revised manuscript. In addressing these comments, we were able to further validate and solidify the conclusions of the study, which we believe substantially strengthened the manuscript. We sincerely appreciate the time and effort dedicated to reviewing our submission and hope that the revisions satisfactorily address all concerns raised.

Reviewer #1 (Remarks to the Author)

Brea and colleagues describe a new genetically engineered mouse model of PCa bearing the concomitant prostate specific loss of *Pten* and *Foxa1*. By coupling classical histopathological studies with approaches of single cell sequencing and spatial transcriptomics, the authors describe a critical oncosuppressive role of *Foxa1* in *Pten*-loss driven mouse prostate tumorigenesis. Their findings highlight two major phenotypes associated with the loss of *Foxa1*: the skewing of the luminal lineage versus the basal/squamous lineage of the prostatic epithelium combined with the generation of an immunosuppressive tumor microenvironment.

The manuscript is well-written, the topic is clinically relevant, and the results are interesting.

Below few major aspects that Authors should carefully consider and address:

1. To better understand the relevance of the genetic model established in mirroring human disease, the Authors should define the incidence of PTEN-FOXA1 double null condition in human prostate cancer considering primary, metastatic and castration resistant stages of the disease.

R: Thank you for this important suggestion. When generating these genetically engineered mouse models, *Pten* deletion served as a general tumor driving event in order to study the role of FOXA1 within a PCa context. We chose *Pten* deletion as our tumor driving event because *Pten* null mice are very commonly used to study PCa and phenotypically recapitulates the human disease well (PMID: 14522255).

To better understand the incidence of PTEN-FOXA1 double null condition in human PCa, we analyzed publicly available datasets containing primary prostate adenocarcinoma or metastatic castration-resistant prostate cancer (CRPC) cases for correlations between PTEN copy-number and FOXA1 mRNA expression level. While we did not observe any significant overall correlation, we observe some cases in which patients with PTEN copy-number loss also exhibit low FOXA1 expression (Quadrant III below), signifying PTEN and FOXA1 loss are not mutually exclusive and the PTEN-FOXA1 low condition is observed in human prostate cancer. This data has been added to **Figure S1A**.

Additionally, adult male mice have been shown to have lower endogenous androgen levels as compared to human, such that mice exhibit similar androgen levels as humans who have undergone androgen ablation therapy (PMCID: PMC4009979). Taken together, *Foxa1* loss in the context of *Pten* deletion in our model represents more advanced PCa with lineage plasticity.

2. Absence of basal cell markers such as TP63 and CK5 is generally coupled to H&E staining in PCa diagnosis. Authors should indicate if and how the expansion of TP63 staining (and maybe also CK5) recapitulates specific genetic or histological subtypes of human PCa.

R: Thank you for this comment. Intraductal prostate adenocarcinoma is typically identified by the presence of an intact basal cell barrier surrounding glandular luminal structures, as observed in our *Pten*-null mice, and in previous studies on *Pten*-null mice. Typically, invasive carcinoma is identified by a loss of p63/KRT5+ basal layer and invasion of luminal epithelial cells into stromal regions (PMID: 28389514). However, in our model, when FOXA1 is deleted in the context of *Pten* KO, we interestingly observed a broad gain in basal gene expression by tumor epithelial cells. Moreover, we observe a gain in squamous differentiation and gene expression. Thus, our model doesn't represent majority of primary PCa, but rather models advanced PCa with lineage plasticity and squamous differentiation. Rather, the gained p63 and squamous differentiation points to the de-differentiation of luminal tumor cells and lineage plasticity upon *Foxa1* loss. Such phenomena of lost luminal identity and increased lineage plasticity can be observed in advance castration-resistant prostate cancer as cells de-differentiate and lose their dependence on AR. Moreover, A previous study analyzing metastatic castration-resistant prostate cancer samples identified a subset of double negative prostate cancer tumors with squamous differentiation. Squamous adenocarcinoma of the prostate, though rare, is associated with treatment resistance and aggressive disease (PMID: 17407588 PMID: 15105655). Thus, we believe our model represents a castration-resistant subtype of prostate cancer with gained lineage plasticity.

3. Figure 1 nicely describes the histology of the different lobes of the mouse prostate in the wild type, *Pten*-null and *Pten-Foxa1* double null model, while 1F defines the lethality of the PF condition compared to P only. Although bladder obstruction may most likely be the reason for lethality, the authors should present a careful analysis of proximal lymph nodes, distant organs (e.g., liver and lung), and bone (e.g., femurs) to exclude or perhaps include metastatic disease progression, which, given the paucity of metastatic GEMMs for PCa, would greatly increase the relevance of the study.

R: Thank you for inquiring about this important point. To address the incidence of metastasis in our novel GEMM models, we examined tissue sections from lymph nodes, lung, and liver by IHC for AR and H&E staining for evidence of metastatic PCa cells. Organs of 9 intact PF mice and 7 intact P mice between the ages of 15-21wks were evaluated, and none were found to exhibit evidence of metastasis. However, we observed evidence of metastasis following castration in XPF mice. 2 out of 5 XPF mice demonstrated evidence of metastasis, one in the lung and one in lymph node tissue, while 0 out of 5 XP mice evaluated exhibited evidence of metastasis by IHC/H&E analysis (as summarized in the table below now found in **Figure S1G**).

Sample Type	Total Mice Evaluated	Mice with Metastasis	Metastasis Incidence (%)
PF (intact)	9	0	0%
P (intact)	7	0	0%
XPF (castrated)	5	2	40%
XP (castrated)	5	0	0%

4. Bulk and scRNAseq studies demonstrate a different immune profile of P and PF prostate tumors with PF described as more immune suppressive/exhausted than P. A critical aspect of these analyses is related to the substantial different stage of P and PF tumors. As clearly described in Figure 1, PF tumors are much more aggressive and advanced than P tumors at the same time points. Therefore, it is difficult, if not impossible, to conclude whether it is the loss of Foxa1 or the stage of the tumor that defines immune suppression/exhaustion. These analyses should be performed on histopathological similar -no age matched- P and PF tumors (i.e., P24 vs PF12).

R: The reviewer brings up a great point. Indeed, it is difficult to determine whether the immune phenotype observed is due to direct regulation by Foxa1 or tumor stage. To address this question, we have compared P18 vs PF12 samples shown below. We observed that PF12 mice still show a greater abundance of macrophages, monocyte-like, CD8 T cell, and CD4 T cells as compared to P18, and that these populations are largely skewed to an immunosuppressive phenotype. Furthermore, it is important to note that Foxa1 loss leads to a histologically distinct phenotype characterized by basal-squamous de-differentiation, which is not observed in Pten-null mice, regardless of timepoint. Thus, an early timepoint PF tumor will not be histologically similar to a late timepoint P tumor.

5. Based on the data shown in Figure 4 (now Figure 5) immune cells appear to share common areas of PF tumor. Are these areas characterized by specific CAFs/extracellular matrix conditions. Fibroblasts are weirdly neglected in this work.

R: CAFs/extracellular matrix conditions are certainly important players in the TME. We observe regions of reactive stroma in the PF mice by histopathology analysis, as discussed in Fig 1. However, in this paper we decided to focus our attention on the immune compartment. We believe the T cell localization in the stromal

regions surrounding tumor epithelial cells reflects its exclusion from properly infiltrating into the tumor cell beds. Nevertheless, we evaluated the proportion of stromal/fibroblast populations in P vs PF scRNA-seq data, and did not find observed consistent marked differences between genotypes (data shown below). An in-depth analysis of what role CAF populations, in particular, may be playing in contributing to the phenotype merits further investigation in future studies, but is beyond the scope of this paper. We will plan to evaluate the relationship among *Foxa1*-null PCa cells and fibroblasts in further depth in follow-up studies.

6. Figure 4 (now Figure 5) describe the spatial distribution of immune cells into the diseased prostate gland of P and PF mice. Are the different percentages described an average of n=? different P and PF tumors or the comparison of just a single P versus a single PF tumors. Are the data consistent across different set (i.e 100 um/10 sections away from those in Figure 4) of sections of the same couple of P and PF tumors?

R: Thank you for these questions. The percentage refers to the percent of total captured area determined to contain that cell type by TESLA analysis, within each sample. (We've now added this description to the Figure legend.) In the Visium spatial transcriptomics data, we are comparing a pair of P12 vs PF12 and P18 vs PF18 mouse tumors. Though our number of replicates for our spatial transcriptomics analysis are low, due to the expense and time required for these experiments (a key limitation of this study), our results between P vs PF genotypes are consistent across timepoints. Moreover, we also performed IHC staining to confirm the observed phenotype, as shown in **Figure 5D**.

7. Figure 5 describes the ability of PF tumors to generate an immune suppressor tumor microenvironment by modulating extracellular signaling. However, the conclusions are completely inferred from the differential RNA expression profiles, which is not sufficient to support the conclusion. Intratumor quantification of the most interesting factors through MS or ELISA methodologies should be included.

R: We thank the reviewers for raising this concern. To further support our scRNA-seq data demonstrating gained TGFB cytokine signaling in PF compared to P tumors, we evaluated our Visium spatial transcriptomics data, as a means of intratumor quantification of TGFB cytokine expression. In accordance with our scRNA-seq data, we observed increased expression of *Tgfb1/2/3* in PF Visium data compared to P (**Figure S6F-G**). To further validate these data at the protein level, we performed IHC staining of PF vs. P tumor tissues for pSMAD2 as an indicator of TGFB signaling. In line with our RNA level data, we observed increased pSMAD2 staining in PF tumors compared to P (**Figure 6G**). Altogether, we have observed increased TGFB signaling in PF tumors compared to P tumors, confirmed by several modalities.

8. As well, the immune suppressor role of specific factors (at least one) should be functionally demonstrated in vivo through the injection of specific inhibitors.

R: We agree it is important to functionally demonstrated the immunosuppressive effect of FOXA1-loss induced cytokine pathways. To this end, we decided to focus on the TGFB pathway, and performed a co-culture assay with WT or FOXA1 KD PC3 prostate cancer cells co-cultured with monocyte derived macrophages (MDMs). Treatment with TGF- β receptor I inhibitor Galunisertib (TGFBI) in this co-culture system led to a reduction in macrophage infiltration, suggesting FOXA1-loss induced TGFB signaling recruit macrophages to the tumor immune microenvironment. This data has now been added to **Figure 6H**.

In addition, to evaluate whether targeting FOXA1-loss induced TGF- β signaling reduced tumor growth *in vivo*, we treated PF tumor-derived allograft mouse models with 75 mg/kg TGFBi twice daily, for three weeks, and found that TGFBi led to a reduction in tumor weight (though not significant). This data has now been added to **Figure S6J**.

9. Interestingly, based on panel B PF tumors express very low levels of Activin (Activin A?) compared to P tumors. Activin A has been described plays a very important role in controlling prostate tissue homeostasis in mouse adult prostate progenitors by promoting quiescence (Cambuli et al., 2022 EMBO Reports). Organoids that were able to spontaneously overcome the barrier by reducing Activin A expression/secretion experience genomic instability and generate CK8/CK5 double positive hyper proliferating dysplastic lesions when transplanted orthotopically into the prostate of adult wild type syngeneic mice. Authors should comment on the possible implication of reduced Activin expression in PF versus P prostate tumors.
R: Thank you for this suggestion. Indeed, activin A plays important roles in prostate tissue homeostasis. However, we only found Activin A downregulated in PF mice at the 18wk timepoint, not the 12wk timepoint. We chose to only discuss pathways found consistently enriched or de-enriched across both 12wk and 18wk timepoints, to support more robust conclusions.
10. A recent paper (De Felice et al., 2024) describes the role of Retinoic Acid signaling in promoting luminal lineage of mouse prostate progenitors by promoting Foxa1 expression. Lack of RA-Foxa1 activity as well as specific mutant forms of FOXA1 associated with human PCa determine AR deregulation and a basal/squamous commitment of prostate progenitor cells. Authors should carefully discuss this paper in their manuscript.
R: Thank you for bringing this recent relevant paper to our attention. We have now added a discussion of this paper to our discussion section.

Reviewer #2 (Remarks to the Author):

FOXA1 is a critical gene in prostate cancer development and therapeutic resistance, with functions that appear to be highly context-dependent. It is frequently mutated and overexpressed in prostate cancer, rarely deleted, but appears to be downregulated in castration-resistant prostate cancer (CRPC). Its primary role is as a pioneer factor for the androgen receptor (AR), and it has been implicated in AR-independent CRPC evolution through mechanisms such as cellular plasticity.

This complexity has led to seemingly conflicting findings, underscoring the multifaceted role that FOXA1 plays in prostate cancer pathogenesis. In this context, the generation and characterization of the PTEN-FOXA1 prostate knockout model presented in this manuscript is of considerable interest. The authors have previously investigated FOXA1 downregulation in human cells, primarily using shRNA, observing reduced cell proliferation in hormone-sensitive lines and upregulation of the epithelial-to-mesenchymal transition (EMT) program.

Interestingly, in this mouse model, co-deletion of PTEN and FOXA1 results in an aggressive prostate cancer phenotype with pronounced basal and squamous features and an immune infiltrate rich in myeloid cells. This suggests a potentially immunosuppressive tumor microenvironment characterized by dysfunctional T cells. Although basal/squamous prostate cancers are relatively rare in patients, I find this model highly intriguing, as it aligns with reports suggesting that this phenotype—more commonly observed in mice—may represent a plastic intermediate state in patients.

The authors employ a multimodal characterization of these tumors, which may have broader implications beyond the specific role of FOXA1. This is a significant strength of the manuscript, and I believe the study would benefit greatly from additional analyses and functional experiments.

Major Points:

Castration Studies

FOXA1 downregulation occurs later in prostate cancer progression, particularly during CRPC evolution, as noted by the authors. Characterizing these models post-castration—whether through surgical means or degarelix administration—is essential to fully demonstrate their relevance. The single-cell RNA sequencing (scRNA-seq) data indicate that the luminal compartment remains substantial. The impact of castration on these cells, as well as on the immune microenvironment, may influence the cancer phenotype and progression.

R: Thank you for this suggestion. It is important to note that intact adult male mice naturally exhibit low androgen levels similar to those observed in patients undergoing standard androgen deprivation therapy (ADT), whereas castrated mice exhibit extremely low androgen levels similar to abiraterone-treated patients (PMID: 23775398). Thus, our intact PF mice exist in a low-androgen environment that resembles the androgen-low conditions under which many post-ADT CRPC tumors arise, wherein FOXA1 is often found downregulated. Castration of PF mice creates an even more androgen-depleted setting, similar to the more advanced CRPC context following treatment with abiraterone. To assess the role of FOXA1 loss under these more severe androgen-deprived conditions, we have now evaluated a cohort of castrated P versus PF mice (abbreviated XP and XPF), as suggested by the reviewer. We found *Foxa1* loss promotes castration resistance in surgically castrated PF vs. P mice. Prostate tumors of castrated PF mice (XPF) show an aggressive, invasive, poorly differentiated and disordered cellular phenotype, with further loss of luminal identity and gained basal-squamous phenotype. Moreover, *Foxa1* depleted tumors show an enhanced proliferative capacity as compared to P tumors following surgical castration, with XPF tumors showing larger tumor size and greater Ki67 staining. These findings have been incorporated into **Figures 1 & 2**, and are also shown below with key data.

12wk post castration, 10X, 40X

12wk post castration, 40X

XP vs XPF Prostate Weight

Castrated

Metastatic Potential

Consistent with their previous studies in human cell lines, the authors report EMT upregulation upon FOXA1 deletion in these models. This raises an important question: do these tumors develop metastases? It is unclear whether the authors have investigated this aspect. Analyzing metastatic spread in both intact and castrated mice would provide a more comprehensive characterization of the model.

R: As described in response to Reviewer 1 Question 3, to address the incidence of metastasis in our novel GEMM models, we examined tissue sections from lymph nodes, lung, and liver by IHC for AR and H&E staining for evidence of metastatic PCa cells. 9 intact PF mice and 7 intact P mice between the ages of 15-21wks were evaluated, and none were found to exhibit evidence of metastasis. However, we observed evidence of metastasis following castration in XPF mice. 2 out of 5 XPF mice demonstrated evidence of metastasis, one in the lung and one in lymph node tissue, while 0 out of 5 XP mice evaluated exhibited evidence of metastasis by IHC/H&E analysis. This information has been added to a summary table in **Figure S1G**.

Addressing Discrepancies in Tumor Immune Microenvironment (TIME) Analysis

The authors provide an extensive characterization of the TIME using multiple single-cell resolution technologies, including scRNA-seq, PIP-seq, Cytek spectral flow cytometry, and spatial transcriptomics. However, while scRNA-seq and spatial transcriptomics reveal prominent macrophage and monocyte infiltration, flow cytometry identifies polymorphonuclear myeloid-derived suppressor cells (PMN-MDSCs) as the dominant immune cell type. PMN-MDSCs are of great interest in prostate cancer, as they play a crucial role in immune suppression in both mouse models and patients. The discrepancy between methods raises concerns. One possible explanation is methodological bias introduced by cell dissociation and tissue preparation. On the other hand, the flow cytometry panel used may not be optimal for distinguishing MDSCs subsets. The standard antibody combination for M-MDSCs and PMN-MDSCs includes Ly6C (HK1.4) and Ly6G (1A8). However, the authors appear to have used Ly6C (HK1.4) in combination with the Ly6G/C antibody (RB6-8C5). When this antibody is used instead of a Ly6G-specific antibody such as 1A8, it stains both M-MDSCs and PMN-MDSCs, resulting in the majority of CD11b-positive cells being positive for this staining, as shown in Fig. S3. This could potentially lead to an overestimation of the number of PMN-MDSCs. Revisiting the staining strategy including additional validation with alternative antibodies is recommended.

R: We thank the reviewer for pointing out the issue regarding the flow cytometry panel for defining MDSC subsets. We agree that the standard antibody combination for M-MDSCs and PMN-MDSCs includes Ly6C (HK1.4) and Ly6G (1A8). We apologize for the incorrect labeling of “Ly6G” as “Ly6G_C” in the original Fig S3. The error has now been corrected. As the percentage of Ly6G⁺ cells (PMN-MDSCs) among CD11b⁺ cells varied across individual samples, a more representative staining panel is now presented now in **Figure S4D**. Additionally, the anti-Ly6G antibody information in Table S3 has been updated from Ly6G/C (RB6-8C5) to Ly6G (1A8).

With regards to the apparent discrepancy between the 10X Genomics scRNA-seq, PIP-seq, flow cytometry, and spatial transcriptomics data, we believe several factors may be at play. First, different tissue dissociation methods and analysis modalities can indeed influence the relative abundance of captured cell types, as shown in PMID: 36750562. Also, discussions with the PIPseq technical support indicated that PIPseq single-cell analysis does not efficiently capture monocytic populations, hence leading to our low recovery of monocytes by PIPseq. Additionally, PMN-MDSCs are known to be poorly captured by scRNA-seq platforms, due to their low RNA content, high RNase activity, and membrane fragility, explaining why they were not represented in our transcriptomics analyses. Lastly, monocytes and MDSCs share several myeloid lineage markers in common and can be challenging to distinguish based solely on RNA expression. We annotated monocyte-like populations based on expression of S100a8, S100a9, and Cd14 genes. However, MDSCs also express these markers. Thus, it is likely this “monocyte” population includes M-MDSCs.

Functional Experiments

The manuscript is largely descriptive. The authors acknowledge in the discussion that targeting cytokines such as TGF- β could influence immune checkpoint inhibitor (ICI) responses. This reasoning could also be extended to the observed TIME components. Performing a cell depletion assay or cytokine-targeting

experiments—even without ICI combination—would provide functional evidence for the importance of at least one of these TIME components.

R: As addressed in Reviewer 1 Question 8, we agree it is important to functionally demonstrated the immunosuppressive effect of FOXA1-loss induced cytokine pathways. To this end, we performed a functional assay with WT or FOXA1-KD PC3 prostate cancer cells co-cultured with monocyte derived macrophages (MDMs) from PCa patients. Treatment with TGFBI in this co-culture system led to a reduction in macrophage infiltration, suggesting FOXA1-loss induced TGFBI signaling recruit macrophages to the tumor immune microenvironment. This data has now been added to **Figure 6H**, also shown below.

Additional Major Considerations:

Throughout the manuscript, including the title and discussion, the authors repeatedly emphasize how their data support the role of FOXA1 in prostate cancer progression through its regulation of prostate epithelial cell lineage identity and the tumor immune microenvironment. While the data may be consistent with this hypothesis, the manuscript lacks direct evidence—either genetic (e.g., lineage tracing) or mechanistic (e.g., ChIP-seq)—to prove it. Alternative explanations cannot be ruled out, making this the weakest aspect of the study.

To address this, I suggest two possible courses of action:

- The authors could revise the text to reduce emphasis on this aspect, ensuring that the conclusions are appropriately cautious.
- If the editors and authors believe this hypothesis is central to the study, additional experiments are necessary to provide stronger evidence.

R: Thank you for this comment. To further support our claim that FOXA1 regulates epithelial lineage identity and inflammatory signaling, we performed ChIP-seq for FOXA1 and H3K27ac. Notably, ChIP-seq analysis revealed FOXA1 chromatin binding at/near lineage and cytokine/inflammatory genes of interest, with corresponding changes in H3K27ac, suggesting their direct transcriptional regulation by FOXA1. This data has now been incorporated into the manuscript in **Figures 3F-H, S3C-E, & S6H**.

Determining the Luminal or Basal Origin of the Tumor

The initial study describing the PB-Cre mice used in this study reported that "there was no evidence of Cre expression in the basal cells of any of the prostate lobes examined" (Wu et al., *Mechanisms of Development*, 2001). However, subsequent studies have shown that this model is not luminal-specific and that Cre is expressed in both basal and luminal epithelial cells (Wang et al., *PNAS*, 2006 [<https://doi.org/10.1073/pnas.051065210>]). Thus, to support the claim that FOXA1 drives dedifferentiation, the authors should investigate the tumor cell of origin in greater detail. The current data do not exclude a basal origin, and the two selected time points (12 and 18 weeks) do not clearly demonstrate a phenotypic switch. To determine whether luminal or basal cells initiate oncogenic transformation in this model, several approaches are available:

- Lineage tracing: This can be achieved using transgenic mouse models with CreERT2 systems more specific to either basal (e.g., CK5-CreERT2, TP63-CreERT2) or luminal (e.g., CK8-CreERT2, NKX3.1-CreERT2) epithelial cells, which would be more appropriate than the PB-Cre model.
- In vitro modeling: Isolating the different prostate epithelial cells (e.g., as organoids), transforming them genetically, and studying their oncogenic potential could provide more direct mechanistic insights. Performing similar experiments in human cells would further strengthen the study.

R: Though PbCre may be expressed in both luminal and basal epithelial cells, FOXA1 is only expressed by luminal epithelial cells. Thus, FOXA1 deletion should only be occurring in luminal cells. Nevertheless, to better support our claim that FOXA1 loss drives de-differentiation of prostate luminal epithelial cells to a basal/squamous phenotype, we evaluated ChIP-seq data of FOXA1 in LNCaP cells and found FOXA1 binding peaks at/near basal/squamous lineage genes such as KRT14 and KRT6A/B/C (GSE55007) (**Figure S3G**). To further understand how FOXA1 loss regulates epithelial lineage in human PCa, we evaluated previously published gene expression microarray analysis of LNCaP cells with FOXA1 KO and rescue with WT or mutant FOXA1 (G87R, D226N, M253K, 254-257FENG>C(del), F266S, L388M) (PMID: 31324884). We found that FOXA1 KO leads to an increase in KRT14 and KRT6A expression. Rescue with WT FOXA1 subsequently repressed KRT14 and KRT6A, while rescue with FOXA1 forkhead mutants, which are known to exhibit reduced chromatin binding, had reduced ability to repress KRT14 and KRT6A expression (**Figure S3F**). Taken together, these findings suggest several squamous genes are direct transcriptional targets of FOXA1 in human PCa.

Chromatin Immunoprecipitation (ChIP)

The authors acknowledge that FOXA1's role is highly context-dependent. While the upregulation of EMT is consistent with previous findings in human cancer cell lines, the observed basal/squamous phenotype and cytokine signaling are novel insights specific to this model. To establish FOXA1's role in this process, ChIP-seq analysis in the relevant cell population is crucial. This would clarify:

- Whether FOXA1 directly regulates luminal differentiation.
- Whether the observed immunosuppressive TIME is a result of direct transcriptional repression of cytokines by FOXA1 or merely a consequence of the basal/squamous PCa phenotype, which may have its own distinct transcriptional network and immune landscape.

R: As suggested, we performed FOXA1 ChIP-seq in P vs. PF prostate tumors to identify specific FOXA1 binding events on the chromatin. We observed FOXA1 chromatin binding at/near luminal and basal/squamous lineage genes of interest, suggesting FOXA1 directly regulates their expression and functions as a master regulator of epithelial lineage in prostate cancer cells. This data is now included in **Figure 3F-H & S3C-E**. We also observe FOXA1 chromatin binding at/near immunosuppressive genes, such as TGFB cytokine genes, now included in **Figure S6H**.

In conclusion, I see significant value in publishing a study describing the characterization of this PTEN/FOXA1 prostate cancer model, which represents a rare but intriguing subtype of prostate cancer observed in patients. However, I would recommend a more comprehensive characterization, including castration of the mice—either surgically or through degarelix treatment—as the genetic alterations in this model are intended to mimic castration-resistant prostate cancer. Additionally, an analysis of potential metastases should be included.

While I understand the inclination to link FOXA1 to cellular plasticity and propose it as a direct regulator of the tumor immune microenvironment through transcriptional control of the chemokine/cytokine milieu, this interpretation remains speculative. Although the characterization may suggest such a role, the study does not provide definitive proof. If the authors wish to pursue this hypothesis—which I agree is of considerable interest—they should conduct a combination of genetic, molecular, and functional analyses to substantiate their claims. Otherwise, they should adjust the manuscript’s tone and conclusions accordingly.

R: We appreciate the reviewers’ careful and thoughtful evaluation of our work and are grateful for the constructive feedback. In response to the comments, we have added substantial experiments and analyses, including expanding our analyses to the castrated condition, evaluating metastasis, and performing ChIP-seq and additional functional experiments, as described in our point-by-point response. We believe these additions significantly strengthen the manuscript and more robustly support our conclusions. We have also revised the title of our manuscript.

Reviewer #3 (Remarks to the Author):

Overview:

This manuscript (ms) explores the tumorigenic effects of Foxa1 depletion in prostate cancer by comparing single-cell and spatial transcriptomic profiles between Pten-null transgenic mice (P) and Pten/Foxa1 double-knockout mice (PF). Comprehensive bioinformatics analyses were conducted using various software tools to support the study's findings. Overall, this manuscript is well-written, with data and analysis results clearly presented. However, several issues need to be addressed.

Major Comments:

1. The study primarily analyzes four samples: P12, PF12, P18, and PF18 — representing two conditions at two different time points. A key limitation of the study is the lack of biological replicates for each combination of condition and time point. This is particularly concerning for the spatial analysis using TESLA, as different tissue structures of replicate samples could lead to varying conclusions.

R: We agree with the reviewers that the lack of biological replicates for our scRNA-seq and spatial transcriptomics analyses, due to their high cost, are a limitation of our study. However, we have confirmed our major conclusions via a variety of modalities, including scRNA-seq, spatial transcriptomics, IHC, and flow cytometry analyses. Moreover, we observe consistent phenotypes across 12wk and 18wk timepoints, further validating the robustness of our findings.

2. In the scRNA-seq integration analysis, there seem to be six samples in total: P12, PF12, P18, PF18, and two castrated samples (XP and XFP). If the two castrated samples are indeed part of the integration analysis (as indicated in the analysis code), their absence from the UMAP visualization is unclear. Furthermore, there is no mention or representation of these castrated samples in any analysis or figures throughout the ms.

R: Thank you for this comment. We've re-incorporated the castrated samples in our UMAPs and manuscript discussions, considering reviewer comments and the important roles of FOXA1 loss in castration-resistant disease. Data on castrated mice has now been incorporated into **Figures 1 & 2**.

3. In the scRNA-seq integration analysis, several cell clusters co-express luminal and basal genes. For instance, the small basal (red) cluster in the center of Fig.2A shows high Krt8 expression in Fig.2E, and similar patterns are seen in the "basal squamous" and "basal proliferating" clusters in Fig.S2A and S2B. Could these clusters represent doublets?

R: Thank you for this comment. Our data processing pipeline includes doublet removal, so it is unlikely these represent doublets. We believe this overlap in luminal and basal markers may suggest a state of lineage plasticity or some basal-luminal intermediate state which is promoted by Foxa1 depletion.

4. What does the UMAP plot in Fig.S2D look like when coloured by sample? Providing this visualization would help clarify sample-specific clustering and potential batch effects.

R: Thank you for this suggestion. As suggested, we have recolored the UMAP plot in Fig. S2D (now **Figure S2F**) by sample, as shown below. Based on this visualization, there does not appear to be any clear batch effects, the samples appear well integrated.

5. In the pseudotime analysis, both the “Basal proliferating” and “Tff3 Lum” clusters are assigned a zero pseudotime (Fig.2D). Does this imply that these clusters were excluded from the Monocle3 pseudotime trajectory analysis? The UMAP also suggests that luminal cells are closer to basal cells than basal proliferating cells, which seems counterintuitive.

R: Thank you for this question. “Basal proliferating” and “Tff3 Lum” clusters were assigned to a separate trajectory path that’s not part of the main trajectory. We utilized Monocle3’s ability to produce separate partitions/trajectories via clustering. Monocle3 does not enforce that all cells must descend from a common transcriptional ancestor and are assigned pseudotime of 0 by default. The basal proliferating cells may have very distinct transcript of cycling genes and may represent a functionally separate pool of cells.

6. How was the starting point chosen for the Monocle3 pseudotime analysis? Have the authors tested the robustness of their results by selecting different starting points — for example, the basal cluster on the left side of the UMAP?

R: Thank you for this question. Yes, we have tested various starting points before choosing this particular one. We tried using the luminal population only (the one we submitted), the Pbsn luminal population, as well as various others (not shown). While choosing Pbsn as the starting point puts the population first, the rest of the trajectory remains similar, with Basal population at the end.

pseudotime analysis starting with Luminal population (shown in manuscript)

pseudotime analysis starting with Pbsn Lum population (not shown in manuscript)

P12 (shown in manuscript)

P12 (not shown in manuscript)

7. The TESLA tool was used to annotate Visium data at the pixel level, but it appears that individual pixels do not have unique cell-type annotations. For example, in Fig.4A (PF18), there is substantial overlap between luminal and basal cells. Does this suggest that both luminal and basal cells occupy the same H&E image pixels? This ambiguity complicates the interpretation of cell-type proportions, as seen in Fig.4B, where macrophages, monocytes, and T cells together constitute over 80% of the tissue — a proportion unlikely to reflect the true composition.

R: Thank you for this comment. Because Visium is not single cell resolution, but rather each barcoded spot may contain 5-10 cells, then each barcoded spot may contain several different cell types, and thus, each spot may express genes corresponding to several different cell types. Thus, TESLA, which annotates Visium data based on a combination of gene expression data and H&E staining, may annotate overlapping regions as containing multiple cell types. The percentages depicted for immune populations in the manuscript should not be interpreted as absolute proportions of the tumor, but rather % of captured spots in the tissue section with high probability of containing the indicated cell type. This clarification has now been added to the figure

legend. With regards to the luminal and basal overlap, we believe the vast overlap of luminal and basal-like regions identified in the PF tumors could be indicative of a mixed luminal-basal phenotype and lineage plasticity, as observed in our scRNA-seq analyses as well.

8. Stromal cell populations are not clearly presented in the Visium profiles. Are stromal cells present in the analyzed tissues, and if so, where are they located?

R: Thank you for this question. Stromal cells are present in the tissues analyzed by Visium. To visualize the location of mesenchymal stromal cells in the analyzed tissue sections, we performed TESLA analysis of the Visium tissue sections based on expression of mesenchymal markers *Vim*, *Col1a1*, *Col3a1*, *Col6a1*, *Col6a2*, *Dcn*, *Tagln*. Notably, P prostate tissue sections appeared to maintain their glandular architecture, with tumor epithelial cell regions surrounded by well-defined mesenchymal stromal cell regions. Meanwhile, in PF tissue sections, TESLA analysis for mesenchymal stromal populations further pointed towards a gain in lineage plasticity in PF tumors, as broad regions of tumor epithelial cells appeared to express these mesenchymal markers. This data has now been added to **Figure S5C**.

Minor Comments:

1. Consistent colours for cell-type granularity across Fig.S2D and S2E would improve visual clarity and aid in the interpretation of results.

R: As suggested, we have recolored the UMAP in **Figure S2F** to match color in the pseudotime plots.

Response to Referees

Manuscript Title: FOXA1 loss drives basal/squamous de-differentiation of prostate cancer and induces an immunosuppressive tumor microenvironment

Manuscript ID: NCOMMS-25-07398A-Z

Dear Reviewers,

We thank you for the continued evaluation of our manuscript and for the helpful feedback provided. In this revision, we have addressed the remaining concerns by further clarifying our bioinformatics methods and confirming the robustness of our findings. Below, we provide a point-by-point response to the comments raised, with all changes highlighted in the revised manuscript.

Reviewer #1 (Remarks to the Author):

Thanks to the authors for addressing the main issues.

The work is now suitable for publication.

R: We thank the reviewer for their careful re-evaluation and positive assessment of the revised manuscript.

Reviewer #2 (Remarks to the Author):

The authors have carefully and thoroughly addressed all of the points raised in my previous review. The vast majority of the additional experiments requested have been performed, and the resulting data strengthen the overall conclusions of the study. Importantly, these new analyses have improved both the clarity and the robustness of the manuscript. I am pleased to see that the manuscript represents a substantial improvement over the original submission.

R: We thank the reviewer for the careful re-evaluation of the manuscript and for the positive and encouraging feedback. We are grateful that the additional experiments and analyses were found to strengthen the conclusions and improve the clarity and robustness of the study.

Reviewer #3 (Remarks to the Author):

I appreciate the authors' efforts in addressing the comments. Most concerns have been adequately addressed. However, several important issues remain.

Major comment #3:

There is no clear description of doublet removal in the data analysis pipeline. The only mention of doublets appears in the TrypLE dissociation step during sample preparation, which does not guarantee the removal of all doublets for the downstream analysis. Proper doublet detection and filtering should be performed computationally during data processing, using established tools such as scDbIFinder or DoubletFinder.

R: Thank you for bringing this question to our attention. Indeed, doublet removal was not performed as part of the data analysis pipeline, as there was no clear consensus on doublet removal at the time this data was collected and analyzed.

In light of these comments and developments in doublet-removal tools, we reprocessed our scRNA-seq data using scDbIFinder, as suggested, to assess its effect on downstream analyses and conclusions. Notably, doublets were found to represent less than 5% of total identified cells in this dataset, and the percentage of doublets was very similar across all samples (**Rebuttal Table 1, Rebuttal Figure 1**). Importantly, overall percentages of cell types/clusters do not change appreciably whether doublets

are removed or included (**Rebuttal Tables 2-4**). Taken together, doublet removal does not appear to have much effect on this dataset and should not affect conclusions drawn by our original analysis.

Rebuttal Fig. 1

Rebuttal Table 1: Percentage of doublets across samples

Sample	% doublets
P12	4.33%
PF12	4.58%
P18	4.89%
PF18	5.05%
XP	4.74%
XPF	5.08%
Overall	4.81%

Rebuttal Table 2: Major cell type percentages with or without doublet removal

Cell types major	singlets	% cell type with doublet removal	singlets + doublets	% cell type without doublet removal
Luminal	7013	38.25%	7537	39.13%
Basal	7066	38.54%	7378	38.30%
Seminal Vesicles	1069	5.83%	1105	5.74%
Stroma	570	3.11%	584	3.03%
Monocytes	1389	7.58%	1406	7.30%
Macrophage	769	4.19%	790	4.10%
B Cells	258	1.41%	260	1.35%
T/NK Cells	201	1.10%	202	1.05%

Rebuttal Table 3: Granular cell type percentages with or without doublet removal

Cell types granular	singlets	% cell type with doublet removal	singlets + doublets	% cell type without doublet removal
Luminal	4639	25.30%	5108	26.52%
Spink1 Lum	1349	7.36%	1361	7.07%
Pbsn Lum	681	3.71%	702	3.64%
Tff3 Lum	344	1.88%	366	1.90%
Seminal Vesicles	1069	5.83%	1105	5.74%
Basal	4683	25.54%	4822	25.03%
Basal Squamous	1893	10.32%	2009	10.43%
Basal Proliferating	490	2.67%	547	2.84%
Fibroblasts	470	2.56%	483	2.51%
Endothelial	100	0.55%	101	0.52%
Monocytes	1389	7.58%	1406	7.30%
Macrophage	769	4.19%	790	4.10%
T/NK cells	201	1.10%	202	1.05%
B cells	258	1.41%	260	1.35%

Rebuttal Table 4: Cluster percentages (resolution 0.2) with or without doublet removal

res0.2 cluster	singlets	% cell type with doublet removal	singlets + doublets	% cell type without doublet removal
0	4639	25.30%	5108	26.52%
1	3322	18.12%	3446	17.89%
2	1893	10.32%	2009	10.43%
3	1389	7.58%	1406	7.30%
4	1349	7.36%	1361	7.07%
5	1219	6.65%	1229	6.38%
6	829	4.52%	855	4.44%
7	769	4.19%	790	4.10%
8	681	3.71%	702	3.64%
9	490	2.67%	547	2.84%
10	470	2.56%	483	2.51%
11	344	1.88%	366	1.90%
12	258	1.41%	260	1.35%
13	240	1.31%	250	1.30%
14	201	1.10%	202	1.05%
15	142	0.77%	147	0.76%
16	100	0.55%	101	0.52%

Continuation of Major comment #3:

The current cell-type annotation is unconvincing. In Fig S2A, two clusters that are far apart from each other on UMAP are both annotated as basal cells (and similarly for the two well-separated Spink1 luminal clusters). According to the code on GitHub, clustering at a resolution of 0.2 yields 17 clusters (0–16). It is unclear why multiple, transcriptionally distant clusters (e.g., clusters 1, 5, and 15) are assigned the same basal cell identity. What marker genes distinguish these clusters (e.g., cluster 1 vs 5)?

Furthermore, the UMAP from the epithelial integration analysis (in the monocle3 analysis) suggests the presence of two transcriptionally distinct basal cell subsets (as well as two distinct Spink1 luminal subsets). These observations indicate meaningful biological heterogeneity that is not sufficiently explained by the current annotations. The rationale for assigning identical cell-type labels to these distinct clusters should be clearly justified, and the annotations revised if necessary.

R: Thank you for this thoughtful comment. Indeed, the tumor epithelium is quite heterogeneous and clusters into several sub-populations (**Rebuttal Fig 2**). However, our focus in our scRNA-seq analysis was to evaluate how *Foxa1* loss dysregulates epithelial cell lineage, signaling, and the tumor immune microenvironment, rather than identifying granular epithelial subclusters. Thus, we combined some epithelial subclusters identified by resolution 0.2 clustering based on canonical cell type marker gene expression to reduce the number of clusters and focus on the main takeaways from this data.

To begin, evaluating the expression of major cell type marker genes across resolution 0.2 clusters (**Rebuttal Fig 3 right panel, also added to Supplementary Figure 2A**), we decided to combine 13 & 6 due to their shared elevated expression of seminal vesicle marker gene *Pate4*, as well as their proximity in the UMAP. Next, cluster 15 was combined with cluster 1 due to similar expression of key basal marker genes and proximity in the UMAP. Cluster 5 was also combined with 1 in the basal population due to the gained expression of key basal genes, which were not expressed in most luminal clusters other than Spink1+ cells, and the reduced expression of luminal markers (**Rebuttal Figure 3 & new Supplementary Figure 2C**). Thus, Cluster 5 might represent an intermediate basal population, which hasn't fully lost luminal gene expression but has begun to gain basal gene expression. This has now been added to Supplementary Figure 2A,C. Moreover, recoloring the pseudotime trajectory by resolution 0.2 clusters, we observed clusters 15 & 5 largely appear within a similar "pseudotime" as cluster 1 (**Rebuttal Fig 4**), supporting their incorporation as part of the "basal" population.

Regarding cluster 4 (the Spink1+ luminal population), though it appears as two physically separate clusters on the UMAP, it was transcriptionally similar enough to be identified as a single cluster at resolution 0.2, likely due to the 2D UMAP visualization not accurately reflecting their global distance. Further, prostate cells that are strongly Spink1+ are well known as a special luminal cell type associated with aggressive prostate cancer with enhanced lineage plasticity, supported by co-expression of luminal and basal genes in our data. Because the focus of this manuscript was not to granularly subtype epithelial populations, we did not further subcluster or explore these populations.

Rebuttal Fig. 2

Rebuttal Fig. 3

Rebuttal Fig 4

Major comment #7:

Regarding the statement "... but rather % of captured spots in the tissue section with high probability of containing the indicated cell type," the definition of "high probability" is unclear. What probability threshold is used to classify a pixel as containing a given cell type? Or what exactly is the criterion for counting pixels for each cell type? While it is correct that Visium spots do not have single-cell resolution and may contain multiple cells, this argument does not apply at the pixel level, as image pixels are substantially smaller than individual cells and should therefore mostly correspond to a single cell type.

R: We thank the reviewer for this thoughtful question. In TESLA analysis, the reported values are not deconvolution-style probabilities of cell-type presence at the pixel level. Instead, as described in the original TESLA publication, the method assigns each pixel a score between 0 and 1 that reflects the relative enrichment of a given cell type at that pixel across the whole slide. To identify regions enriched for a specific cell type, pixels with enrichment scores >0.3 were classified as enriched for that cell type and used to quantify the corresponding area. Notably, because these scores represent relative

enrichment rather than exclusive assignment, a pixel may have non-zero scores for multiple cell types, capturing their colocalization pattern.

The advantages of using the score from TESLA over spot-level deconvolution are threefold:

1. TESLA incorporates H&E image information, providing additional morphological context to improving cell-type annotation compared with deconvolution approaches based solely on spot-level gene expression.
2. TESLA enables annotation across the entire tissue section, rather than being restricted to measured Visium spots, which typically cover < 50% of the captured tissue area.
3. Pixel-level scores provide improved visualization and quantification of spatial co-localization among different cell types compared with spot-level analysis.

We have clarified this point in the methods and legends of the revised manuscript to avoid confusion.

Additional point related to Major comment #7:

In the statement "... we believe the vast overlap of luminal and basal-like regions identified in the PF tumours could be indicative of a mixed luminal–basal phenotype and lineage plasticity, as observed in our scRNA-seq analyses as well," it is unclear which specific scRNA-seq clusters in the integration analysis represent this "mixed luminal–basal phenotype." The authors should explicitly identify the relevant clusters and provide supporting evidence for this interpretation.

R: We thank the reviewer for this question and opportunity to clarify this statement. This statement here was not made in reference to a particular scRNA-seq cluster, but rather a trend we observed of gained basal phenotype among PF tumor epithelial cells, without complete loss of luminal gene expression, suggesting a "mixed" identity or lineage plasticity, which was supported by several pieces of evidence from our scRNA-seq analysis. Firstly, evaluation of *Krt5* (basal), *Krt6a* (squamous), and *Krt8* (luminal) expression on the UMAP for PF tumors showed evidence of epithelial cells with gained basal/squamous gene expression but limited loss of luminal marker *Krt8* expression (Figure 2E). Additionally, our lineage trajectory analysis revealed that PF tumors exhibit a much earlier gain in basal gene expression along the trajectory, with more cells within the "early" luminal populations expressing basal marker genes, as compared to P tumors (Figure 2G). Moreover, we observed increased expression of epithelial-mesenchymal transition (EMT) genes among epithelial cells in PF tumors (Fig S2F & Fig 3A), further supporting *Foxa1* loss induces lineage plasticity. Nevertheless, as described in response to comment #3 above, further evaluation of canonical luminal and basal cell type marker genes across resolution 0.2 clusters suggests cluster 5 may represent a mixed or intermediate basal state.

Minor comment: It might be better to include Fig S2C in Fig 2D.

R: We have now included FigS2C in Fig2D, as suggested.

Reviewer #3 (Remarks on code availability):

The code on the GitHub is well organized and easy to follow.

R: Thank you.

Response to Referees

We confirm that there are no additional reviewer comments to address.

We sincerely thank the reviewers and the editors for their careful evaluation of our manuscript and for their thoughtful and constructive feedback. Their insights have significantly strengthened our work, and we greatly appreciate the time and expertise invested in the review process.